# NOISE BALANCE AND STATIONARY DISTRIBUTION OF STOCHASTIC GRADIENT DESCENT

## ABSTRACT

The stationary distribution of a stochastic system often provides fundamental insights into its dynamics, yet the stationary distribution of the stochastic gradient descent (SGD) algorithm, a cornerstone of machine learning, remains analytically elusive. In this work, we first show that the minibatch noise of SGD regularizes the solution towards a noise-balanced, low-dimensional subspace when the loss function exhibits rescaling symmetry. This result allows us to construct a type of linear network that captures the depth and width and for which a general stationary distribution of the stochastic gradient flow can be derived. This stationary distribution reveals complex nonlinear phenomena, including phase transitions, loss of ergodicity, memory effects, sign coherence, and fluctuation inversion. These phenomena are shown to exist uniquely in deep networks, highlighting a fundamental distinction between deep and shallow models. Finally, we discuss the implications of our proposed theory for practical problems in variational Bayesian inference.

## 1 INTRODUCTION

In both natural and social sciences, the stationary distribution of a stochastic system often holds the key to understanding the underlying dynamics of complex processes (Van Kampen, 1992; Rolski et al., 2009). For the stochastic gradient descent (SGD) algorithm, a foundational tool in modern machine learning, understanding its stationary distribution has the potential to provide deep insights into its learning behavior. However, despite extensive use, the stationary distribution of SGD remains analytically elusive. The stochastic gradient descent (SGD) algorithm is defined as $\Delta\theta_t = -\frac{\eta}{S}\sum_{x\in B}\nabla_\theta\ell(\theta, x)$ where $\theta$ is the model parameter and $\ell(\theta, x)$ is a per-sample loss whose expectation over $x$ gives the training loss: $L(\theta) = \mathbb{E}_x[\ell(\theta, x)]$. $B$ is a randomly sampled minibatch of data points, each independently sampled from the training set, and $S$ is the minibatch size. In this work, we adopt the SDE approximation of SGD (Latz, 2021; Li et al., 2019; 2021; Sirignano & Spiliopoulos, 2020; Fontaine et al., 2021; Hu et al., 2017):

$$d\theta = -\nabla_\theta L dt + \sqrt{TC(\theta)}dW_t, \qquad (1)$$

where $C(\theta) = \mathbb{E}[\nabla\ell(\theta)\nabla^T\ell(\theta)] - \mathbb{E}[\nabla\ell(\theta)]\mathbb{E}[\nabla^T\ell(\theta)]$ is the gradient covariance, $dW_t$ is a stochastic process satisfying $dW_t \sim N(0, Idt)$ and $\mathbb{E}[dW_t dW_{t'}^T] = \delta(t - t')I$, and $T = \eta/S$. Apparently, $T$ gives the average noise level in the dynamics. Previous works have suggested that the ratio $T$ is a main factor determining the behavior of SGD, and using different $T$ often leads to different generalization performance (Shirish Keskar et al., 2016; Liu et al., 2021; Ziyin et al., 2022b).

Our main contributions are

1. the derivation of the "law of balance," which shows that SGD converges to a low-dimensional subspace when the rescaling symmetry is present in the loss function (Section 3);
2. identification of a minimal linear model with the concepts of width and depth, for which a general form of the stationary distribution of SGD is found (Section 4);
3. discovery of novel phenomena in this model such as phase transitions, sign coherence (impossibility to learn the wrong sign), loss of ergodicity, memory effects, and fluctuation inversion (Section 4).

The next section discusses the closely related works, especially on known examples of stationary distributions of SGD. All proofs and derivations are given in Appendix A.

## 2 RELATED WORKS

**Stationary distribution of SGD**. The FP equation is a high-dimensional partial differential equation whose solution (and its existence) is an open problem in mathematics and many fields of sciences and only known for a few celebrated special cases (Risken & Risken, 1996). One of the earliest works that computes the stationary distribution of SGD is the Lemma 20 of Chaudhari & Soatto (2018), which assumes that the noise has a constant covariance and shows that if the loss function is quadratic, then the stationary distribution is Gaussian. Similarly, using a Laplace approximation expansion and assuming that the noise is parameter-independent, a series of recent works showed that the stationary distribution of SGD is exponential in the model parameters close to a local minimum: $p(\theta) \propto \exp[-a\theta^T H\theta]$, for some constant $a$ and matrix $H$ (Mandt et al., 2017; Xie et al., 2020; Liu et al., 2021). Assuming that the noise covariance only depends on the loss function value $L(\theta)$, Mori et al. (2022) and Wojtowytsch (2024) showed that the stationary distribution is power-law-like and proportional to $L(\theta)^{-c_0}$ for some constant $c_0$. A primary feature of these previous results is that stationary distribution does not exhibit any memory effect and also preserves ergodicity. Until now, the general form of the analytical solution to the stationary distribution of SGD is unknown.

**Minimal Solvable Models.** With the exact noise covariance, only two minimal models have been solved exactly. Ziyin et al. (2021) solved a minimal model when the loss function is of the form $\ell(w) = (xw^2 - y)^2$, and found that the solution takes the form:

$$P(w) \propto (w^2 + S\sigma^2)^{-1-Sa/2\eta+S^2b\sigma^2/\eta} \exp\left(-\frac{Sbw^2}{\eta}\right), \qquad (2)$$

where $a = \mathbb{E}[xy], b = \mathbb{E}[x^2]$ are determined from the input data and $\sigma$ represents the strength of the additive noise. While this model exhibits an interesting phase transition from escaping the saddle to convergence to the saddle, it has no notion of width and depth. A more recent example is solved in Chen et al. (2023) where the loss function takes the form of $\ell'(W_1, W_2) = \|W_2W_1x - y\|^2$ with $W$ being a matrix. Under special assumptions on the initialization (balanced), data distribution (isotropic), and label noise (structured), the authors showed that the dynamics SGD reduces to a similar problem with width 1: $\ell = (uwx - y)^2$, and defining $v = uw$, the stationary distribution is found to be

$$P(v) \propto \exp\left(-\frac{v}{\eta\sigma^2}\right) v^{-\frac{3}{2}-\frac{v}{\eta\sigma^2}}. \qquad (3)$$

This model is more advanced than Eq. 2 but still lacks the notion of depth and has a trivial dependence on the width. Additionally, this solution is a particular solution that only applies to a special initialization, and it is unclear whether this is the actual distribution found by SGD.

**Symmetry and SGD dynamics**. Also related to our work is the study of how symmetry affects the learning dynamics of SGD (Kunin et al., 2020). A closely related work is (Li et al., 2020), which studies the dynamics of SGD when there is scale invariance, conjecturing that SGD reaches a fast equilibrium state at the early stage of training. Our result is different as we study a different type of symmetry, the rescaling symmetry.

## 3 NOISE BALANCE UNDER THE RESCALING SYMMETRY

We show that when the loss function exhibits the rescaling symmetry, SGD will evolve towards a solution for which the gradient noise is balanced. In Section 4, we will leverage this result to construct a model for which the stationary distribution of SGD only has support on a very low-dimensional subspace.

### 3.1 RESCALING SYMMETRY AND LAW OF BALANCE

A type of invariance – the rescaling symmetry – often appears in the loss function and it is preserved for all sampling of minibatches. The per-sample loss $\ell$ is said to have the rescaling symmetry for all $x$ if $\ell(u, w, x) = \ell(\lambda u, w/\lambda, x)$ for a scalar $\lambda \in \mathbb{R}_+$. This type of symmetry appears in many scenarios in deep learning. For example, it appears in any neural network with the ReLU activation. It also appears in the self-attention of transformers, often in the form of key and query matrices (Vaswani et al., 2017). When this symmetry exists between $u$ and $w$, one can prove the following result, which we refer to as the law of balance.

**Theorem 3.1.** *Let $u$, $w$, and $v$ be parameters of arbitrary dimensions. Let $\ell(u,w,v,x)$ satisfy $\ell(u,w,v,x) = \ell(\lambda u, w/\lambda, v, x)$ for arbitrary $x$ and any $\lambda \in \mathbb{R}_+$. Then,*

$$\frac{d}{dt}(\|u\|^2 - \|w\|^2) = -T(u^T C_1 u - w^T C_2 w), \tag{4}$$

*where $C_1 = \mathbb{E}[A^T A] - \mathbb{E}[A^T]\mathbb{E}[A]$, $C_2 = \mathbb{E}[AA^T] - \mathbb{E}[A]\mathbb{E}[A^T]$ and $A_{ki} = \partial\tilde{\ell}/\partial(u_i w_k)$ with $\tilde{\ell}(u_i w_k, v, x) \equiv \ell(u_i, w_k, v, x)$. In addition, if Eq. 4 does not vanish, there exists a unique $\lambda^* \in \mathbb{R}_+ \bigcup\{\infty\}$ such that $\dot{C} = 0$ if the training proceeds with $\ell(\lambda^* u, w/\lambda^*, v, x)$.*

*Remark* 3.2. A key step in the derivation is that the Brownian motion term vanishes in the time-evolution of $\|u\|^2 - \|w\|^2$. Therefore, the time evolution of this quantity follows an ODE rather than an SDE. Essentially, this is because when there is symmetry in the loss, the gradient noise is low-rank.

Equation (4) will be referred to as the law of balance. Here, $v$ stands for the parameters that are irrelevant to the symmetry, and $C_1$ and $C_2$ are positive semi-definite by definition. The theorem still applies if the model has parameters other than $u$ and $w$. The theorem can be applied recursively when multiple rescaling symmetries exist. See Figure 1 for an illustration the the dynamics and how it differs from other types of GD. While the matrices $C_1$ and $C_2$ may not always be full-rank, the law of balance is often well-defined and gives nontrivial result. Below, we prove that in a quite general setting, for all *active* hidden neurons of a two-layer ReLU net, $C_1$ and $C_2$ are full-rank (Theorem 3.3).

The law of balance implies two different types of balance. The first type of balance is the balance of gradient noise. The proof of the theorem shows that the stationary point of the law in (4) is equivalent to

$$\text{Tr}_w[C(w)] = \text{Tr}_u[C(u)], \tag{5}$$

where $C(w)$ and $C(u)$ are the gradient covariance of $w$ and $u$, respectively. Therefore, SGD prefers a solution where the gradient noise between the two layers is balanced. Also, this implies that the balance conditions of the law is only dependent on the diagonal terms of the Fisher information (if we regard the loss as a log probability), which is often well-behaved.[1]

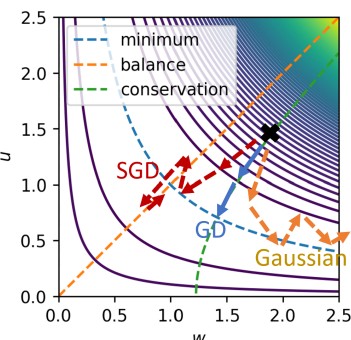

Figure 1: Dynamics of GD and SGD and GD with injected Gaussian noise for the simple problem $\ell(u,w) = (uwx - y)^2$. Due to the rescaling symmetry between $u$ and $w$, GD follows a conservation law: $u^2(t) - w^2(t) = u^2(0) - w^2(0)$, SGD converges to the balanced solution $u^2 = w^2$, while GD with injected noise diverges due to simple diffusion in the degenerate directions.

The second type of balance is the norm ratio balance between layers. Equation (4) implies that in the degenerate direction of the rescaling symmetry, a single and unique point is favored by SGD. Let $u = \lambda u^*$ and $w = \lambda^{-1} w^*$ for arbitrary $u^*$ and $w^*$, then, the stationary point of the law is reached at $\lambda^4 = \frac{(w^*)^T C_2 w^*}{(u^*)^T C_1 u^*}$. The quantity $\lambda$ can be called the "balancedness" of the norm, and the law states that when a rescaling symmetry exists, a special balancedness is preferred by the SGD algorithm. When $C_1$ or $C_2$ vanishes, $\lambda$ or $\lambda^{-1}$ diverges, and so does SGD. Therefore, having a nonvanishing noise actually implies that SGD training will be more stable. For common problems, $C_1$ and $C_2$ are positive definite and, thus, if we know the spectrum of $C_1$ and $C_2$ at the end of training, we can estimate a rough norm ratio at convergence:

$$-T(\lambda_{1M}\|u\|^2 - \lambda_{2m}\|w\|^2) \leq \frac{d}{dt}(\|u\|^2 - \|w\|^2) \leq -T(\lambda_{1m}\|u\|^2 - \lambda_{2M}\|w\|^2),$$

where $\lambda_{1m(2m)}$ and $\lambda_{1M(2M)}$ represent the minimal and maximal eigenvalue of the matrix $C_{1(2)}$, respectively. Thefore, the value of $\|u\|^2/\|w\|^2$ is restricted by (See Section A.5)

$$\frac{\lambda_{2m}}{\lambda_{1M}} \leq \frac{\|u\|^2}{\|w\|^2} \leq \frac{\lambda_{2M}}{\lambda_{1m}}. \tag{6}$$

---

[1]That the noise will balance does not imply that either trace will converge or stay close to a fixed value – it is also possible for both terms to oscillate while their difference is close to zero.

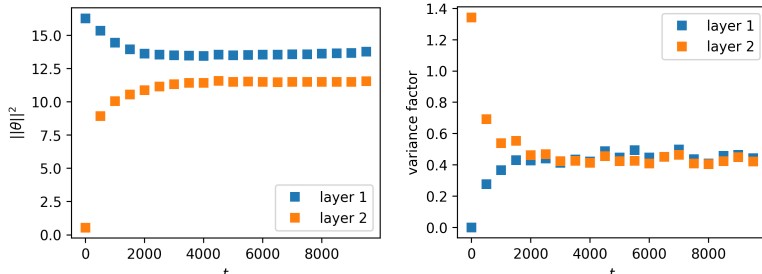

Figure 2: A two-layer ReLU network trained on a full-rank dataset. **Left**: because of the rescaling symmetry, the norms of the two layers are balanced approximately (but not exactly). **Right**: the first and second terms in Eq. (4). We see that both terms evolve towards a point where they exactly balance. In agreement with our theory, SGD training leads to an approximate norm balance and exact gradient noise balance.

Thus, a remaining question is whether the quantities $u^T C_1 u$ and $w^T C_2 w$ are generally well-defined and nonvanishing or not. The following proposition shows that for a generic two-layer ReLU net, $u^T C_1 u$ and $w^T C_2 w$ are almost everywhere strictly positive. We define a two-layer ReLU net as

$$f(x) = \sum_i^d u_i \text{ReLU}(w_i^T x + b_i), \tag{7}$$

where $u_i \in \mathbb{R}^{d_u}, w_i \in \mathbb{R}^{d_w}$ and $b_i$ is a scalar with $i$ being the index of the hidden neuron. For each $i$, the model has the rescaling symmetry: $u_i \to \lambda u_i, (w_i, b_i) \to (\lambda^{-1} w_i, \lambda^{-1} b_i)$. We thus apply the law of balance to each neuron separately. The per-sample loss function is

$$\ell(\theta, x) = \|f(x) - y(x, \epsilon)\|^2. \tag{8}$$

Here, $x$ has a full-rank covariance $\Sigma_x$, and $y = g(x) + \epsilon$ for some function $g$ and $\epsilon$ is a zero-mean random vector independent of $x$ and have the full-rank covariance $\Sigma_\epsilon$. The following theorem shows that for this network, $C_1$ and $C_2$ are full rank unless the neuron is "dead".

**Theorem 3.3.** *Let the loss function be given in Eq. (8). Let $C_1^{(i)}$ and $C_2^{(i)}$ denote the corresponding noise matrices of the $i$-th neuron, and $p_i := \mathbb{P}(w_i^T x + b_i > 0)$. Then, $C_1^{(i)}$ and $C_2^{(i)}$ are full-rank for all $i$ such that $p_i > 0$.*

See Figure 2. We train a two-layer ReLU network with the number of neurons: $20 \to 200 \to 20$. The dataset is a synthetic data set, where $x$ is drawn from a normal distribution, and the labels: $y = x + \epsilon$, for an independent Gaussian noise $\epsilon$ with unit variance. While every neuron has a rescaling symmetry, we focus on the overall rescaling symmetry between the two weight matrices. The norm between the two layers reach a state of approximate balance – but not a precise balance. At the same time, the model evolves during training towards a state where $u^T C_1 u$ and $w^T C_2 w$ are balanced.

Standard analysis shows that the difference between SGD and GD is of order $T^2$ per unit time step, and it is thus often believed that SGD can be understood perturbatively through GD (Hu et al., 2017). However, the law of balance implies that the difference between GD and SGD is not perturbative. As long as there is any level of noise, the difference between GD and SGD at stationarity is $O(1)$. This theorem also implies the loss of ergodicity, an important phenomenon in nonequilibrium physics (Palmer, 1982; Thirumalai & Mountain, 1993; Mauro et al., 2007; Turner et al., 2018), because not all solutions with the same training loss will be accessed by SGD with equal probability.

### 3.2 1D RESCALING SYMMETRY

The theorem greatly simplifies when both $u$ and $w$ are one-dimensional.

**Corollary 3.4.** *If $u, w \in \mathbb{R}$, then, $\frac{d}{dt}|u^2 - w^2| = -T C_0 |u^2 - w^2|$, where $C_0 = \text{Var}[\frac{\partial \ell}{\partial(uw)}]$.*

Before we apply the theorem to study the stationary distributions, we stress the importance of this balance condition. This relation is closely related to Noether's theorem (Misawa, 1988; Baez & Fong, 2013; Malinowska & Ammi, 2014). If there is no weight decay or stochasticity in training, the quantity $\|u\|^2 - \|w\|^2$ will be a conserved quantity under gradient flow (Du et al., 2018; Kunin et al., 2020; Tanaka & Kunin, 2021), as is evident by taking the infinite $S$ limit. The fact that it monotonically decays to zero at a finite $T$ may be a manifestation of some underlying fundamental

mechanism. A more recent result in Wang et al. (2022) showed that for a two-layer linear network, the norms of two layers are within a distance of order $O(\eta^{-1})$, suggesting that the norm of the two layers are balanced. Our result agrees with Wang et al. (2022) in this case, but our result is stronger because our result is nonperturbative, only relies on the rescaling symmetry, and is independent of the loss function or architecture of the model. It is useful to note that when $L_2$ regularization with strength $\gamma$ is present, the rate of decay changes from $TC_0$ to $TC_0 + \gamma$. This points to a nice interpretation that when rescaling symmetry is present, the implicit bias of SGD is equivalent to weight decay. See Figure 1 for an illustration of this point.

This reveals a fundamental difference between the SGD gradient noise and the full-rank Langevin noise that happens in nature. See Figure 3, where we run SGD on the simple loss function $\ell = (uwx - y)^2$ for $x \in \mathbb{R}$ drawn from a Gaussian distribution, and $y = x + \epsilon$. The lack of fluctuation for the quantity $|u^2 - w^2|$ is consistent with the theory that the noise vanishes in this subspace.

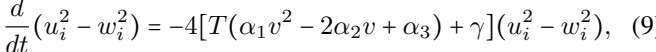

**Example: two-layer linear network.** It is instructive to illustrate the application of the law to a two-layer linear network, the simplest model that obeys the law. Let $\theta = (w, u)$ denote the set of trainable parameters; the per-sample loss is $\ell(\theta, x) = (\sum_j^d u_i w_i x - y)^2 + \gamma\|\theta\|^2$. Here, $d$ is the width of the model, $\gamma\|\theta\|^2$ is the $L_2$ regularization term with strength $\gamma \geq 0$, and $\mathbb{E}_x$ denotes the averaging over the training set, which could be a continuous distribution or a discrete sum of delta distributions. It will be convenient for us also to define the shorthand: $v := \sum_i^d u_i w_i$. The distribution of $v$ is said to be the distribution of the "model." Applying the law of balance, we obtain that

Figure 3: SGD converges to a balanced solution. The quantity $u^2 - w^2$ is conserved for GD without noise Du et al. (2018), is divergent for GD with an isotropic Gaussian noise, which simulates the simple Langevin model, and decays to zero for SGD, making a sharp and dramatic contrast.

$$\frac{d}{dt}(u_i^2 - w_i^2) = -4[T(\alpha_1 v^2 - 2\alpha_2 v + \alpha_3) + \gamma](u_i^2 - w_i^2), \quad (9)$$

where we have introduced the parameters

$$\alpha_1 := \text{Var}[x^2], \quad \alpha_2 := \mathbb{E}[x^3 y] - \mathbb{E}[x^2]\mathbb{E}[xy], \quad \alpha_3 := \text{Var}[xy]. \quad (10)$$

When $\alpha_1\alpha_3 - \alpha_2^2$ or $\gamma > 0$, the time evolution of $|u^2 - w^2|$ can be upper-bounded by an exponentially decreasing function in time: $|u_i^2 - w_i^2|(t) < |u_i^2 - w_i^2|(0)\exp\left(-4T(\alpha_1\alpha_3 - \alpha_2^2)t/\alpha_1 - 4\gamma t\right) \to 0$. Namely, the quantity $(u_i^2 - w_i^2)$ decays to 0 with probability 1. We thus have $u_i^2 = w_i^2$ for all $i \in \{1, \cdots, d\}$ at stationarity, in agreement with the Corollary.

## 4 STATIONARY DISTRIBUTION OF AN ANALYTICAL MODEL

The law of balance allows us to construct a minimal analytical model that has notions of depth and width, for which the stationary distribution can be found. While linear networks are limited in expressivity, their loss landscape and dynamics are highly nonlinear and exhibits many shared phenomenon with nonlinear neural networks (Kawaguchi, 2016; Saxe et al., 2013). Let $\theta$ follow the high-dimensional Wiener process given by Eq.(1). The probability density evolves according to its Kolmogorov forward (Fokker-Planck) equation:

$$\frac{\partial}{\partial t}p(\theta, t) = -\sum_i \frac{\partial}{\partial \theta_i}\left(p(\theta, t)\frac{\partial}{\partial \theta_i}L(\theta)\right) + \frac{1}{2}\sum_{i,j}\frac{\partial^2}{\partial \theta_i \partial \theta_j}C_{ij}(\theta)p(\theta, t). \quad (11)$$

The solution of this partial differential equation is an open problem for almost all high-dimensional problems. This section solves it for a high-dimensional non-quadratic loss function.

### 4.1 DEPTH-0 CASE

Let us first derive the stationary distribution of a one-dimensional linear regressor, which will be a basis for comparison to help us understand what is unique about having a "depth." The per-sample loss is $\ell(x, v) = (vx - y)^2 + \gamma v^2$. Defining

$$\beta_1 := \mathbb{E}[x^2], \quad \beta_2 := \mathbb{E}[xy], \quad (12)$$

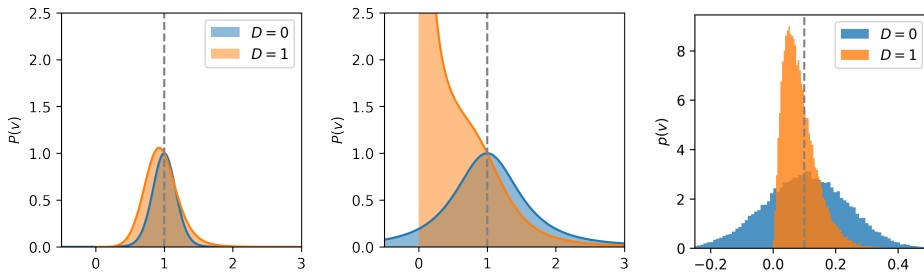

Figure 4: Stationary distributions of SGD for simple linear regression ($D = 0$), and a two-layer network ($D = 1$) across different $T = \eta/S$: $T = 0.05$ (**left**) and $T = 0.5$ (**Mid**). We see that for $D = 1$, the stationary distribution is strongly affected by the choice of the learning rate. In contrast, for $D = 0$, the stationary distribution is also centered at the global minimizer of the loss function, and the choice of the learning rate only affects the thickness of the tail. **Right**: the stationary distribution of a one-layer $\tanh$-model, $f(x) = \tanh(vx)$ ($D = 0$) and a two-layer tanh-model $f(x) = w\tanh(ux)$ ($D = 1$). For $D = 1$, we define $v := wu$. The vertical line shows the ground truth. The deeper model never learns the wrong sign of $wu$ ("sign coherence"), whereas the shallow model can learn the wrong one.

the global minimizer of the loss can be written as: $v^* = \beta_2/\beta_1'$. The gradient variance is also not trivial: $C(v) := \mathrm{Var}[\nabla_v\ell(v,x)] = 4(\alpha_1 v^2 - 2\alpha_2 v + \alpha_3)$, where $\alpha$ is defined in Eq. (10). Note that the loss landscape $L$ only depends on $\beta_1$ and $\beta_2$, and the gradient noise only depends on $\alpha_1$, $\alpha_2$ and, $\alpha_3$. It is thus reasonable to call $\beta$ the landscape parameters and $\alpha$ the noise parameters. Both $\beta$ and $\alpha$ appear in all stationary distributions, implying that the stationary distributions of SGD are strongly data-dependent. Another relevant quantity is $\Delta := \alpha_1 \min_v C(v)/4 \geq 0$, which is the minimal level of noise on the landscape. It turns out that the stationary distribution is qualitatively different for $\Delta = 0$ and for $\Delta > 0$. For all the examples in this work,

$$\Delta = \mathrm{Var}[x^2]\mathrm{Var}[xy] - \mathrm{cov}(x^2, xy) = \alpha_1\alpha_3 - \alpha_2^2. \tag{13}$$

$\Delta$ zero when, for all samples of $(x, y)$, $xy + c = kx^2$ for some constant $k$ and $c$. We focus on the case $\Delta > 0$ in the main text, which is most likely the case for practical situations. The other cases are dealt with in Section A.

For $\Delta > 0$, the stationary distribution for linear regression is (Section A)

$$p(v) \propto (\alpha_1 v^2 - 2\alpha_2 v + \alpha_3)^{-1-\frac{\beta_1'}{2T\alpha_1}} \exp\left[-\frac{1}{T}\frac{\alpha_2\beta_1' - \alpha_1\beta_2}{\alpha_1\sqrt{\Delta}}\arctan\left(\frac{\alpha_1 v - \alpha_2}{\sqrt{\Delta}}\right)\right], \tag{14}$$

in agreement with the previous result in Mori et al. (2022). Two notable features exist for this distribution: (1) the power exponent for the tail of the distribution depends on the learning rate and batch size, and (2) the integral of $p(v)$ converges for an arbitrary learning rate. On the one hand, this implies that increasing the learning rate alone cannot introduce new phases of learning to a linear regression; on the other hand, it implies that the expected error is divergent as one increases the learning rate (or the feature variation), which happens at $T = \beta_1'/\alpha_1$. We will see that deeper models differ from the single-layer model in these two crucial aspects.

## 4.2 An Analytical Model

Now, we consider the following model with a notion of depth and width; its loss function is

$$\ell = \left[\sum_i^{d_0}\left(\prod_{k=0}^D u_i^{(k)}\right)x - y\right]^2, \tag{15}$$

where $D$ can be regarded as the depth and $d_0$ the width. When the width $d_0 = 1$, the law of balance is sufficient to solve the model. When $d_0 > 1$, we need to eliminate additional degrees of freedom. We note that this model conceptually resembles (but not identical to) a diagonal linear network, which has been found to well approximate the dynamics of real networks (Pesme et al., 2021; Nacson et al., 2022; Berthier, 2023; Even et al., 2023).

We introduce $v_i := \prod_{k=0}^D u_i^{(k)}$, and so $v = \sum_i v_i$, where we call $v_i$ a "subnetwork" and $v$ the "model." The following proposition shows that independent of $d_0$ and $D$, the dynamics of this model can be reduced to a one-dimensional form by invoking the law of balance.

**Theorem 4.1.** *For all $i \neq j$, one (or more) of the following conditions holds for all trajectories at stationarity: (1) $v_i = 0$, or $v_j = 0$, or $L(\theta) = 0$; (2) $\mathrm{sgn}(v_i) = \mathrm{sgn}(v_j)$. In addition, (2a) if $D = 1$, for a constant $c_0$, $\log|v_i| - \log|v_j| = c_0$; (2b) if $D > 1$, $|v_i|^2 - |v_j|^2 = 0$.*

This theorem contains many interesting aspects. First of all, the three situations in item 1 directly tell us the distribution of $v$ if the initial state of of $v$ is given by these conditions.[2] This implies a memory effect, namely, that the stationary distribution of SGD can depend on its initial state. The second aspect is the case of item 2, which we will solve below. Item 2 of the theorem implies that all the $v_i$ of the model must be of the same sign for any network with $D \geq 1$. Namely, no subnetwork of the original network can learn an incorrect sign. This is dramatically different from the case of $D = 0$. We will refer to this phenomenon as the "sign coherence" of deep networks. Figure 4 shows an example of this effect in a nonlinear network. The third interesting aspect of the theorem is that it implies that the dynamics of SGD is qualitatively different for different depths of the model. In particular, $D = 1$ and $D > 1$ have entirely different dynamics. For $D = 1$, the ratio between every pair of $v_i$ and $v_j$ is a conserved quantity. In sharp contrast, for $D > 1$, the distance between different $v_i$ is no longer conserved but decays to zero. Therefore, a new balancing condition emerges as we increase the depth. Conceptually, this qualitative distinction also consistent with the result in Ziyin et al. (2022a), where $D = 1$ models are found to be qualitatively different from models with $D > 1$.

With this theorem, we are ready to solve the stationary distribution. It suffices to condition on the event that $v_i$ does not converge to zero. Let us suppose that there are $d$ nonzero $v_i$ that obey item 2 of Theorem 4.1 and $d$ can be seen as an effective width of the model. We stress that the effective width $d \leq d_0$ depends on the initialization and can be arbitrary.[3] Therefore, we condition on a fixed value of $d$ to solve for the stationary distribution of $v$ (Appendix A):

**Theorem 4.2.** *Let $\delta(x)$ denote the Dirac delta function. For an arbitrary factor $z$ $in[0,1]$, an invariant solution of the Fokker-Planck Equation is $p^*(v) = (1-z)\delta(v) + zp_\pm(v)$, where*

$$p_\pm(|v|) \propto \frac{1}{|v|^{3(1-1/(D+1))}g_\mp(v)} \exp\left(-\frac{1}{T}\int_0^{|v|} d|v| \frac{d^{1-2/(D+1)}(\beta_1|v| \mp \beta_2)}{(D+1)|v|^{2D/(D+1)}g_\mp(v)}\right), \quad (16)$$

*where $p_-$ is the distribution on $(-\infty, 0)$ and $p_+$ is that on $(0, \infty)$, and $g_\mp(v) = \alpha_1|v|^2 \mp 2\alpha_2|v| + \alpha_3$.*

The arbitrariness of the scalar $z$ is due to the memory effect of SGD – if all parameters are initialized at zero, they will remain there with probability 1. This means that the stationary distribution is not unique. Since the result is symmetric in the sign of $\beta_2 = \mathbb{E}[xy]$, we assume that $\mathbb{E}[xy] > 0$ from now on. Also, we focus on the case $\gamma = 0$ in the main text.[4]

### 4.3    NONEQUILIBRIUM PHASE TRANSITION AT A CRITICAL $T_c$

The distribution of $v$ is

$$p_\pm(|v|) \propto \frac{|v|^{\pm\beta_2/2\alpha_3 T - 3/2}}{(\alpha_1|v|^2 \mp 2\alpha_2|v| + \alpha_3)^{1\pm\beta_2/4T\alpha_3}} \exp\left(-\frac{1}{2T}\frac{\alpha_3\beta_1 - \alpha_2\beta_2}{\alpha_3\sqrt{\Delta}} \arctan\frac{\alpha_1|v| \mp \alpha_2}{\sqrt{\Delta}}\right). \quad (17)$$

This measure is worth a close examination. First, the exponential term is upper and lower bounded and well-behaved in all situations. In contrast, the polynomial term becomes dominant both at infinity and close to zero. When $v < 0$, the distribution is a delta function at zero: $p(v) = \delta(v)$. To see this, note that the term $v^{-\beta_2/2\alpha_3 T - 3/2}$ integrates to give $v^{-\beta_2/2\alpha_3 T - 1/2}$ close to the origin, which is infinite. Away from the origin, the integral is finite. This signals that the only possible stationary distribution has a zero measure for $v \neq 0$. The stationary distribution is thus a delta distribution, meaning that if $x$ and $y$ are positively correlated, the learned subnets $v_i$ can never be negative, independent of the initial configuration.

For $v > 0$, the distribution is nontrivial. Close to $v = 0$, the distribution is dominated by $v^{\beta_2/2\alpha_3 T - 3/2}$, which integrates to $v^{\beta_2/2\alpha_3 T - 1/2}$. It is only finite below a critical $T_c = \beta_2/\alpha_3$. This is a phase-transition-like behavior. As $T \to (\beta_2/\alpha_3)_-$, the integral diverges and tends to a delta distribution.

---

[2] $L \to 0$ is only possible when $\Delta = 0$ *and* $v = \beta_2/\beta_1$.

[3] One can initialize the parameters such that $d$ takes any value between 1 and $d_0$. One way to achieve this is to initialize on the stationary points specified by Theorem 4.1 at the desired $d$.

[4] When weight decay is present, the stationary distribution is the same, except that one needs to replace $\beta_2$ with $\beta_2 - \gamma$. Other cases are also studied in detail in Appendix A and listed in Table. 1.

Namely, if $T > T_c$, we have $u_i = w_i = 0$ for all $i$ with probability 1, and no learning can happen. If $T < T_c$, the stationary distribution has a finite variance, and learning may happen. In the more general setting, where weight decay is present, this critical $T$ shifts to $T_c = \frac{\beta_2 - \gamma}{\alpha_3}$. When $T = 0$, the phase transition occurs at $\beta_2 = \gamma$, in agreement with the threshold weight decay identified in Ziyin & Ueda (2022). See Figure 4 for illustrations of the distribution across different values of $T$. We also compare with the stationary distribution of a depth-0 model. Two characteristics of the two-layer model appear rather striking: (1) the solution becomes a delta distribution at the sparse solution $u = w = 0$ at a large learning rate; (2) the two-layer model never learns the incorrect sign ($v$ is always non-negative). Another exotic phenomenon implied by the result is what we call the "fluctuation inversion." Naively, the variance of model parameters should increase as we increase $T$, which is the noise level in SGD. However, for the distribution we derived, the variance of $v$ and $u$ both decrease to zero as we increase $T$: injecting noise makes the model fluctuation vanish. We discuss more about this "fluctuation inversion" in the next section.

Also, while there is no other phase-transition behavior below $T_c$, there is still an interesting crossover behavior in the distribution of the parameters as we change the learning rate. When training a model, The most likely parameter we obtain is given by the maximum likelihood estimator of the distribution, $\hat{v} := \arg\max p(v)$. Understanding how $\hat{v}(T)$ changes as a function of $T$ is crucial. This quantity also exhibits nontrivial crossover behaviors at critical values of $T$.

When $T < T_c$, a nonzero maximizer for $p(v)$ must satisfy

$$v^* = -\frac{\beta_1 - 10\alpha_2 T - \sqrt{(\beta_1 - 10\alpha_2 T)^2 + 28\alpha_1 T(\beta_2 - 3\alpha_3 T)}}{14\alpha_1 T}. \tag{18}$$

The existence of this solution is nontrivial, which we analyze in Appendix A.8. When $T \to 0$, a solution exists and is given by $v = \beta_2/\beta_1$, which does not depend on the learning rate or noise $C$. Note that $\beta_2/\beta_1$ is also the minimum point of $L(u_i, w_i)$. This means that SGD is only a consistent estimator of the local minima in deep learning in the vanishing learning rate limit. How biased is SGD at a finite learning rate? Two limits can be computed. For a small learning rate, the leading order correction to the solution is $v = \frac{\beta_2}{\beta_1} + \left( \frac{10\alpha_2\beta_2}{\beta_1^2} - \frac{7\alpha_1\beta_2^2}{\beta_1^3} - \frac{3\alpha_3}{\beta_1} \right) T$. This implies that the common Bayesian analysis that relies on a Laplace expansion of the loss fluctuation around a local minimum is improper. The fact that the stationary distribution of SGD is very far away from the Bayesian posterior also implies that SGD is only a good Bayesian sampler at a small learning rate.

**Example**. It is instructive to consider an example of a structured dataset: $y = kx + \epsilon$, where $x \sim \mathcal{N}(0, 1)$ and the noise $\epsilon$ obeys $\epsilon \sim \mathcal{N}(0, \sigma^2)$. We let $\gamma = 0$ for simplicity. If $\sigma^2 > \frac{8}{21}k^2$, there exists a transitional learning rate: $T^* = \frac{4k + \sqrt{42}\sigma}{4(21\sigma^2 - 8k^2)}$. Obviously, $T_c/3 < T^*$. One can characterize the learning of SGD by comparing $T$ with $T_c$ and $T^*$. For this example, SGD can be classified into roughly 5 different regimes. See Figure 5.

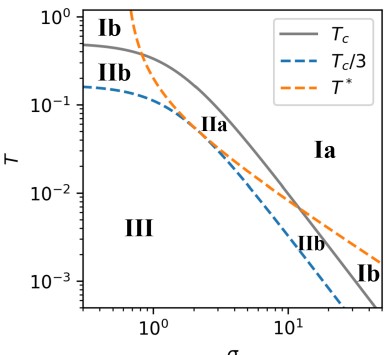

Figure 5: Regimes of learning for SGD as a function of $T$ and the noise in the dataset $\sigma$. According to (1) whether the sparse transition has happened, (2) whether a nontrivial maximum probability estimator exists, and (3) whether the sparse solution is a maximum probability estimator, the learning of SGD can be characterized into 5 regimes. Regime **I** is where SGD converges to a sparse solution with zero variance. In regime **II**, the stationary distribution has a finite spread, but the probability of being close to the sparse solution is very high. In regime **III**, the probability density of the sparse solution is zero, and therefore the model will learn without much problem. In regime **b**, a local nontrivial probability maximum exists. The only maximum probability estimator in regime **a** is the sparse solution.

### 4.4 POWER-LAW TAIL OF DEEPER MODELS

An interesting aspect of the depth-1 model is that its distribution is independent of the width $d$ of the model. This is not true for a deep model, as seen from Eq. (16). The $d$-dependent term vanishes only if $D = 1$. Another intriguing aspect of the depth-1 distribution is that its tail is independent of any hyperparameter of the problem, dramatically different from the linear regression case. This is true for deeper models as well.

Since $d$ only affects the non-polynomial part of the distribution, the stationary distribution scales as $p(v) \propto \frac{1}{v^{3(1-1/(D+1))}(\alpha_1 v^2 - 2\alpha_2 v + \alpha_3)}$. Hence, when $v \to \infty$, the scaling behaviour is $v^{-5+3/(D+1)}$. The tail gets monotonically thinner as one increases the depth. For $D = 1$, the exponent is $7/2$; an infinite-depth network has an exponent of $5$. Therefore, the tail of the model distribution only depends on the depth and is independent of the data or details of training, unlike the depth-0 model. In addition, due to the scaling $v^{5-3/(D+1)}$ for $v \to \infty$, we can see that $\mathbb{E}[v^2]$ will not diverge no matter how large the $T$ is.

An intriguing feature of this model is that the model with at least one hidden layer will never have a divergent training loss. This directly explains the puzzling observation of the edge-of-stability phenomenon in deep learning: SGD training often gives a neural network a solution where a slight increment of the learning rate will cause discrete-time instability and divergence (Wu et al., 2018; Cohen et al., 2021). These solutions, quite surprisingly, exhibit low training and testing loss values even when the learning rate is right at the critical learning rate of instability. This observation contradicts naive theoretical expectations. Let $\eta_{\mathrm{sta}}$ denote the largest stable learning rate. Close to a local minimum, one can expand the loss function up to the second order to show that the value of the loss function $L$ is proportional to $\mathrm{Tr}[\Sigma]$. However, $\Sigma \propto 1/(\eta_{\mathrm{sta}} - \eta)$ should be a very large value (Yaida, 2018; Liu et al., 2021), and therefore $L$ should diverge. Thus, the edge of stability phenomenon is incompatible with the naive expectation up to the second order, as pointed out by Damian et al. (2022). Our theory offers a direct explanation of why the divergence of loss does not happen: for deeper models, the fluctuation of model parameters decreases as the gradient noise level increases, reaching a

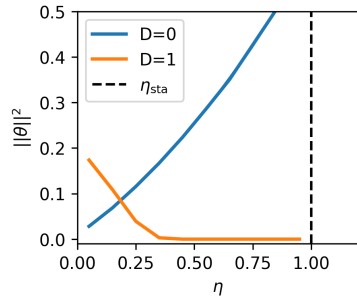

Figure 6: Training loss of a tanh network. $D = 0$ is the case where only the input weight is trained, and $D = 1$ is the case where both input and output layers are trained. For $D = 0$, the model norm increases as the model loses stability. For $D = 1$, a "fluctuation inversion" effect appears. The fluctuation of the model vanishes before it loses stability.

minimal value before losing stability. Thus, SGD has a finite loss because of the power-law tail and fluctuation inversion. See Figure 6.

**Infinite-$D$ limit.** As $D$ tends to infinity, the distribution becomes

$$p(v) \propto \frac{1}{v^{3+k_1}(\alpha_1 v^2 - 2\alpha_2 v + \alpha_3)^{1-k_1/2}} \exp\left(-\frac{d}{DT}\left(\frac{\beta_2}{\alpha_3 v} + \frac{\alpha_2 \alpha_3 \beta_1 - 2\alpha_2^2 \beta_2 + \alpha_1 \alpha_3 \beta_2}{\alpha_3^2 \sqrt{\Delta}} \arctan(\frac{\alpha_1 v - \alpha_2}{\sqrt{\Delta}})\right)\right),$$

where $k_1 = d(\alpha_3 \beta_1 - 2\alpha_2 \beta_2)/(TD\alpha_3^2)$. An interesting feature is that the architecture ratio $d/D$ always appears simultaneously with $1/T$. This implies that for a sufficiently deep neural network, the ratio $D/d$ also becomes proportional to the strength of the noise. Since we know that $T = \eta/S$ determines the performance of SGD, our result thus shows an extended scaling law of training: $\frac{d}{D}\frac{S}{\eta} = const$. The architecture aspect of the scaling law also agrees with an alternative analysis (Hanin, 2018; Hanin & Rolnick, 2018), where the optimal architecture is found to have a constant ratio of $d/D$. See Figure 7.

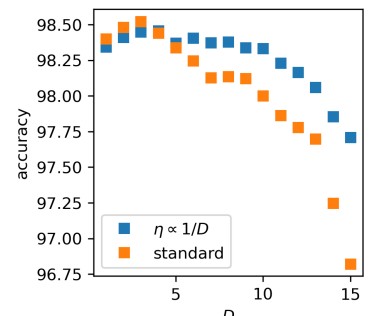

Figure 7: Performance of fully connected tanh nets on MNIST for different depths $D$. The training proceeds with standard Adam. Scaling the learning rate as $1/D$ keeps the model performance relatively unchanged.

Now, if we fix $T$, there are three situations: (1) $d = o(D)$, (2) $d = c_0 D$ for a constant $c_0$, (3) $d = \Omega(D)$. If $d = o(D)$, $k_1 \to 0$ and the distribution converges to $p(v) \propto v^{-3}(\alpha_1 v^2 - 2\alpha_2 v + \alpha_3)^{-1}$, which is a delta distribution at $0$. Namely, if the width is far smaller than the depth, the model will collapse to zero. Therefore, we should increase the model width as we increase the depth. In the second case, $d/D$ is a constant and can thus be absorbed into the definition of $T$ and is the only limit where we obtain a nontrivial distribution with a finite spread. If $d = \Omega(D)$, the distribution becomes a delta distribution at the global minimum of the loss landscape, $p(v) = \delta(v - \beta_2/\beta_1)$ and achieves the global minimum.

### 4.5 VARIATIONAL BAYESIAN LEARNING

An implications of the analytical solution we found is the inappropriateness of using SGD to approximate a Bayesian posterior. Because every SGD iteration can be regarded as a sampling of the model parameters. A series of recent works have argued that the stationary distribution can be used as an approximation of the Bayesian posterior for fast variational inference (Mandt et al., 2017; Chaudhari & Soatto, 2018), $p_{\text{Bayes}}(\theta) \approx p_{\text{SGD}}(\theta)$, a method that has been used for a wide variety of applications (Jospin et al., 2022). However, our result implies that such an approximation is likely to fail. Common in Bayesian deep learning, we interpret the per-sample loss as the log probability and the weight decay as a Gaussian prior over the parameters, the true model parameters have a log probability of

$$\log p_{\text{Bayes}}(\theta|x) \propto \ell(\theta, x) + \gamma \|\theta\|^2. \tag{19}$$

This distribution has a nonzero measure everywhere for any differentiable loss. However, the distribution for SGD in Eq.(16) has a zero probability density almost everywhere because a 1d subspace has a zero probability measure in a high-dimensional space. This implies that the KL divergence between the two distributions (either $\text{KL}(p_{\text{Bayes}} \| p_{\text{SGD}})$ or $\text{KL}(p_{\text{SGD}} \| p_{\text{Bayes}})$) is infinite.

## 5 CONCLUSION

In this work, we established that SGD systematically converges toward a balanced solution when rescaling symmetry is present, a principle we termed the "law of balance." This finding implies that SGD inherently focuses on a low-dimensional subspace in the stationary stage of training, offering new insights into its behavior in deep learning. By leveraging the law of balance, we constructed an analytically solvable model incorporating the concepts of depth and width and successfully derived the stationary distribution of SGD. This analytical solution revealed several previously unknown phenomena, which may have significant implications for understanding deep learning dynamics. One key consequence of our theory is that using SGD to approximate the Bayesian posterior may be fundamentally inappropriate when symmetries exist in the model, a concern particularly relevant for overparameterized models (Nguyen, 2019). For those seeking to employ SGD for variational inference, it may be necessary to eliminate symmetries from the loss function, which presents an intriguing avenue for future research. While our theory provides valuable insights, it is currently limited to a minimal model, and exploring more complex and realistic models will be an essential direction for future studies.

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

## A  THEORETICAL CONSIDERATIONS

### A.1  BACKGROUND

#### A.1.1  ITO'S LEMMA

Let us consider the following stochastic differential equation (SDE) for a Wiener process $W(t)$:

$$dX_t = \mu_t dt + \sigma_t dW(t). \tag{20}$$

We are interested in the dynamics of a generic function of $X_t$. Let $Y_t = f(t, X_t)$; Ito's lemma states that the SDE for the new variable is

$$df(t, X_t) = \left( \frac{\partial f}{\partial t} + \mu_t \frac{\partial f}{\partial X_t} + \frac{\sigma_t^2}{2} \frac{\partial^2 f}{\partial X_t^2} \right) dt + \sigma_t \frac{\partial f}{\partial x} dW(t). \tag{21}$$

Let us take the variable $Y_t = X_t^2$ as an example. Then the SDE is

$$dY_t = \left( 2\mu_t X_t + \sigma_t^2 \right) dt + 2\sigma_t X_t dW(t). \tag{22}$$

Let us consider another example. Let two variables $X_t$ and $Y_t$ follow

$$dX_t = \mu_t dt + \sigma_t dW(t),$$
$$dY_t = \lambda_t dt + \phi_t dW(t). \tag{23}$$

The SDE of $X_t Y_t$ is given by

$$d(X_t Y_t) = (\mu_t Y_t + \lambda_t X_t + \sigma_t \phi_t) dt + (\sigma_t Y_t + \phi_t X_t) dW(t). \tag{24}$$

#### A.1.2  FOKKER PLANCK EQUATION

The general SDE of a 1d variable $X$ is given by:

$$dX = -\mu(X)dt + B(X)dW(t). \tag{25}$$

The time evolution of the probability density $P(x, t)$ is given by the Fokker-Planck equation:

$$\frac{\partial P(X, t)}{\partial t} = -\frac{\partial}{\partial X} J(X, t), \tag{26}$$

where $J(X, t) = \mu(X)P(X, t) + \frac{1}{2} \frac{\partial}{\partial X}[B^2(X)P(X, t)]$. The stationary distribution satisfying $\partial P(X, t)/\partial t = 0$ is

$$P(X) \propto \frac{1}{B^2(X)} \exp\left[ -\int dX \frac{2\mu(X)}{B^2(X)} \right] \coloneqq \tilde{P}(X), \tag{27}$$

which gives a solution as a Boltzmann-type distribution if $B$ is a constant. We will apply Eq. (27) to determine the stationary distributions in the following sections.

### A.2  PROOF OF THEOREM 3.1

*Proof.* We first prove the law of balance, and then prove the uniqueness of $\lambda$.

(**Part I**) We omit writing $v$ in the argument unless necessary. By definition of the symmetry $\ell(\mathbf{u}, \mathbf{w}, x) = \ell(\lambda \mathbf{u}, \mathbf{w}/\lambda, x)$, we obtain its infinitesimal transformation $\ell(\mathbf{u}, \mathbf{w}, x) = \ell((1 + \epsilon)\mathbf{u}, (1 - \epsilon)\mathbf{w}/\lambda, x)$. Expanding this to first order in $\epsilon$, we obtain

$$\sum_i u_i \frac{\partial \ell}{\partial u_i} = \sum_j w_j \frac{\partial \ell}{\partial w_j}. \tag{28}$$

The equations of motion are

$$\frac{du_i}{dt} = -\frac{\partial \ell}{\partial u_i}, \tag{29}$$

$$\frac{dw_j}{dt} = -\frac{\partial \ell}{\partial w_j}. \tag{30}$$

Using Ito's lemma, we can find the equations governing the evolutions of $u_i^2$ and $w_j^2$:

$$\frac{du_i^2}{dt} = 2u_i \frac{du_i}{dt} + \frac{(du_i)^2}{dt} = -2u_i \frac{\partial \ell}{\partial u_i} + T C_i^u,$$

$$\frac{dw_j^2}{dt} = 2w_j \frac{dw_j}{dt} + \frac{(dw_j)^2}{dt} = -2w_j \frac{\partial \ell}{\partial w_j} + T C_j^w, \tag{31}$$

where $C_i^u = \mathrm{Var}\left[\frac{\partial \ell}{\partial u_i}\right]$ and $C_j^w = \mathrm{Var}\left[\frac{\partial \ell}{\partial w_j}\right]$. With Eq. (28), we obtain

$$\frac{d}{dt}(\|u\|^2 - \|w\|^2) = -T(\sum_j C_j^w - \sum_i C_i^u) = -T\left(\sum_j \mathrm{Var}\left[\frac{\partial \ell}{\partial w_j}\right] - \sum_i \mathrm{Var}\left[\frac{\partial \ell}{\partial u_i}\right]\right). \tag{32}$$

Due to the rescaling symmetry, the loss function can be considered as a function of the matrix $uw^T$. Here we define a new loss function as $\tilde{\ell}(u_i w_j) = \ell(u_i, w_j)$. Hence, we have

$$\frac{\partial \ell}{\partial w_j} = \sum_i u_i \frac{\partial \tilde{\ell}}{\partial (u_i w_j)}, \frac{\partial \ell}{\partial u_i} = \sum_j w_j \frac{\partial \tilde{\ell}}{\partial (u_i w_j)}. \tag{33}$$

We can rewrite Eq. (32) into[5]

$$\frac{d}{dt}(\|u\|^2 - \|w\|^2) = -T(u^T C_1 u - w^T C_2 w), , \tag{34}$$

where

$$(C_1)_{ij} = \mathbb{E}\left[\sum_k \frac{\partial \tilde{\ell}}{\partial (u_i w_k)} \frac{\partial \tilde{\ell}}{\partial (u_j w_k)}\right] - \sum_k \mathbb{E}\left[\frac{\partial \tilde{\ell}}{\partial (u_i w_k)}\right]\mathbb{E}\left[\frac{\partial \tilde{\ell}}{\partial (u_j w_k)}\right],$$

$$\equiv \mathbb{E}[A^T A] - \mathbb{E}[A^T]\mathbb{E}[A] \tag{35}$$

$$(C_2)_{kl} = \mathbb{E}\left[\sum_i \frac{\partial \tilde{\ell}}{\partial (u_i w_k)} \frac{\partial \tilde{\ell}}{\partial (u_i w_l)}\right] - \sum_i \mathbb{E}\left[\frac{\partial \tilde{\ell}}{\partial (u_i w_k)}\right]\mathbb{E}\left[\frac{\partial \tilde{\ell}}{\partial (u_i w_l)}\right]$$

$$\equiv \mathbb{E}[A A^T] - \mathbb{E}[A]\mathbb{E}[A^T], \tag{36}$$

where

$$(A)_{ik} \equiv \frac{\partial \tilde{\ell}}{\partial (u_i w_k)}. \tag{37}$$

(**Part II**) The rescaling transformation can be rewritten as

$$\theta_\lambda = A(\lambda)\theta, \tag{38}$$

where

$$A := \begin{pmatrix} \lambda I_u & O \\ O & \lambda^{-1} I_w \end{pmatrix}, \tag{39}$$

and we denote $\theta := (u^T, w^T)^T$. The covariance matrix as a function of $\theta_\lambda$ is given by

$$C(\theta_\lambda) = \mathbb{E}\left[\frac{\partial \ell}{\partial \theta_\lambda}\left(\frac{\partial \ell}{\partial \theta_\lambda}\right)^T\right] - \mathbb{E}\left[\frac{\partial \ell}{\partial \theta_\lambda}\right]\mathbb{E}\left[\frac{\partial \ell}{\partial \theta_\lambda}\right]^T = A^{-1}(\lambda)C(\theta)A(\lambda)^{-1}. \tag{40}$$

---

[5]Alternatively, we provide a conventional proof that explicitly invokes Ito's lemma. By defining $\theta := (u^T, w^T)^T$ and $B := \begin{pmatrix} I_u & O \\ O & -I_w \end{pmatrix}$, the quantity $\|u\|^2 - \|w\|^2$ can be rewritten as $\theta^T B \theta =: D(\theta)$. Using the Ito's lemma (22), the dynamics of $\theta^T B \theta$ can be written as $\dot{D}(\theta) = -\theta^T B \nabla_\theta L + \theta^T B \sqrt{TC(\theta)} dW/dt + T\mathrm{Tr}[C(\theta)B]$. Meanwhile, the infinitesimal form of the rescaling symmetry can be expressed as $\ell(\theta, x) = \ell(A\theta, x)$ with $A := \begin{pmatrix} I_u & O \\ O & -I_w \end{pmatrix}$. We can expand the equation to first order in $\epsilon$ and obtain $\theta^T B \nabla_\theta \ell = 0$. Taking the average to the both sides, we have $\theta^T B \nabla_\theta L = 0$. In addition, we have $\theta^T B C(\theta) = \mathbb{E}[\theta^T B \nabla_\theta \ell \nabla_\theta^T \ell] - \mathbb{E}[\theta^T B \nabla_\theta \ell]\mathbb{E}[\nabla_\theta^T \ell] = 0$. Therefore, $\theta^T B \sqrt{C(\theta)} = 0$ since $C(\theta)$ and $\sqrt{C(\theta)}$ share the same null space. Plugging $\theta^T B \nabla_\theta L = 0$ and $\theta^T B \sqrt{C(\theta)} = 0$ into the evolution equation of $D(\theta)$, we obtain $\dot{D}(\theta) = T\mathrm{Tr}[C(\theta)B]$, which is the same as Eq. (34).

Here, we denote $I_{u(w)}$ as an identity matrix with the dimension $d_u(d_w)$, which is the dimension of the vector $u(w)$. Then,

$$\text{Tr}[C(\theta_\lambda)B] = \text{Tr}[A^{-2}C(\theta)B], \tag{41}$$

where $B := \begin{pmatrix} I_u & O \\ O & -I_w \end{pmatrix}$ is considered as the generator of the transformation matrix $A$ and hence $[A, B] = 0$. The conserved quantity $Q(u, w) = \|u\|^2 - \|w\|^2$ can be also expressed as

$$Q(\theta) = \theta^T B \theta. \tag{42}$$

Then, under the rescaling transformation, we have

$$Q(\theta_\lambda) = \theta^T A^{-1} B A^{-1} \theta = \theta^T \tilde{B} \theta, \tag{43}$$

where

$$\tilde{B} = \begin{pmatrix} \lambda^{-2} I_u & O \\ O & -\lambda^2 I_w \end{pmatrix}. \tag{44}$$

The matrix $\tilde{B}$ can be further decomposed into two subspaces: the subspace with positive eigenvalues(the subspace associated with the vector $u$) and the subspace with negative eigenvalues(the subspace associated with the vector $w$). The function $G(\theta_\lambda) := \dot{C}(\theta_\lambda)$ takes the form of

$$G(\theta_\lambda) = -\gamma(\|u_\lambda\|^2 - \|w_\lambda\|^2) - T(u_\lambda^T C_1 u_\lambda - w_\lambda^T C_2 w_\lambda) = -\gamma \theta_\lambda^T B \theta_\lambda + T\text{Tr}[C(\theta_\lambda)B] \tag{45}$$

in the presence of the weight-decay term. By this decomposition, we obtain

$$G(\theta_\lambda) = -\gamma \theta_\lambda^T B \theta_\lambda + T\text{Tr}[C(\theta_\lambda)B]$$

$$= \frac{1}{\lambda^2} \left( T \sum_{i=1}^{d_u} \text{Var}\left( \frac{\partial \ell}{\partial u_i} \right) + \gamma \|w\|^2 \right) - \lambda^2 \left( T \sum_{j=1}^{d_w} \text{Var}\left( \frac{\partial \ell}{\partial w_j} \right) + \gamma \|u\|^2 \right). \tag{46}$$

Here we assume $\gamma \geq 0$. We can also similarly obtain the case with $\gamma < 0$. Notice that the first term on the right-hand side of Eq. (46) is propotional to $\lambda^{-2}$. Therefore, the unique $|\lambda^*|$ with $G(\theta_\lambda) = 0$ is given by

$$\lambda^* = \left( \frac{T \sum_{i=1}^{d_u} \text{Var}(\partial \ell / \partial u_i) + \gamma \|w\|^2}{T \sum_{j=1}^{d_w} \text{Var}(\partial \ell / \partial w_j) + \gamma \|u\|^2} \right)^{1/4}, \tag{47}$$

which is unique. The proof is complete. $\qquad \square$

### A.3 SECOND-ORDER LAW OF BALANCE

Considering the modified loss function:

$$\ell_{\text{tot}} = \ell + \frac{1}{4} T \|\nabla L\|^2. \tag{48}$$

In this case, the Langevin equations become

$$dw_j = -\frac{\partial \ell}{\partial w_j} dt - \frac{1}{4} T \frac{\partial \|\nabla L\|^2}{\partial w_j}, \tag{49}$$

$$du_i = -- \frac{\partial \ell}{\partial u_i} dt - \frac{1}{4} T \frac{\partial \|\nabla L\|^2}{\partial u_i}. \tag{50}$$

Hence, the modified SDEs of $u_i^2$ and $w_j^2$ can be rewritten as

$$\frac{du_i^2}{dt} = 2u_i \frac{du_i}{dt} + \frac{(du_i)^2}{dt} = -2u_i \frac{\partial \ell}{\partial u_i} + + TC_i^u - \frac{1}{2} Tu_i \nabla_{u_i} |\nabla L|^2, \tag{51}$$

$$\frac{dw_j^2}{dt} = 2w_j \frac{dw_j}{dt} + \frac{(dw_j)^2}{dt} = -2w_j \frac{\partial \ell}{\partial w_j} + TC_j^w - \frac{1}{2} Tw_j \nabla_{w_j} |\nabla L|^2. \tag{52}$$

In this section, we consider the effects brought by the last term in Eqs. (51) and (52). From the infinitesimal transformation of the rescaling symmetry:

$$\sum_j w_j \frac{\partial \ell}{\partial w_j} = \sum_i u_i \frac{\partial \ell}{\partial u_i}, \tag{53}$$

we take the derivative of both sides of the equation and obtain

$$\frac{\partial L}{\partial u_i} + \sum_j u_j \frac{\partial^2 L}{\partial u_i \partial u_j} = \sum_j w_j \frac{\partial^2 L}{\partial u_i \partial w_j}, \tag{54}$$

$$\sum_j u_j \frac{\partial^2 L}{\partial w_i \partial u_j} = \frac{\partial L}{\partial w_i} + \sum_j w_j \frac{\partial^2 L}{\partial w_i \partial w_j}, \tag{55}$$

where we take the expectation to $\ell$ at the same time. By substituting these equations into Eqs. (51) and (52), we obtain

$$\frac{d\|u\|^2}{dt} - \frac{d\|w\|\|^2}{dt} = T \sum_i (C_i^u + (\nabla_{u_i} L)^2) - T \sum_j (C_j^w + (\nabla_{w_j} L)^2). \tag{56}$$

Then following the procedure in Appendix. A.2, we can rewrite Eq. (56) as

$$\frac{d\|u\|^2}{dt} - \frac{d\|w\|^2}{dt} = -T(u^T C_1 u + u^T D_1 u - w^T C_2 w - w^T D_2 w)$$

$$= -T(u^T E_1 u - w^T E_2 w), \tag{57}$$

where

$$(D_1)_{ij} = \sum_k \mathbb{E}\left[\frac{\partial \ell}{\partial(u_i w_k)}\right] \mathbb{E}\left[\frac{\partial \ell}{\partial(u_j w_k)}\right], \tag{58}$$

$$(D_2)_{kl} = \sum_i \mathbb{E}\left[\frac{\partial \ell}{\partial(u_i w_k)}\right] \mathbb{E}\left[\frac{\partial \ell}{\partial(u_i w_l)}\right], \tag{59}$$

$$(E_1)_{ij} = \mathbb{E}\left[\sum_k \frac{\partial \ell}{\partial(u_i w_k)} \frac{\partial \ell}{\partial(u_j w_k)}\right], \tag{60}$$

$$(E_2)_{kl} = \mathbb{E}\left[\sum_i \frac{\partial \ell}{\partial(u_i w_k)} \frac{\partial \ell}{\partial(u_i w_l)}\right]. \tag{61}$$

For one-dimensional parameters $u, w$, Eq. (57) is reduced to

$$\frac{d}{dt}(u^2 - w^2) = -\mathbb{E}\left[\left(\frac{\partial \ell}{\partial(uw)}\right)^2\right](u^2 - w^2). \tag{62}$$

Therefore, we can see this loss modification increases the speed of convergence. Now, we move to the stationary distribution of the parameter $v$. At the stationarity, if $u_i = -w_i$, we also have the distribution $P(v) = \delta(v)$ like before. However, when $u_i = w_i$, we have

$$\frac{dv}{dt} = -4v(\beta_1 v - \beta_2) + 4Tv(\alpha_1 v^2 - 2\alpha_2 v + \alpha_3) - 4\beta_1^2 Tv(\beta_1 v - \beta_2)(3\beta_1 v - \beta_2) + 4v\sqrt{T(\alpha_1 v^2 - 2\alpha_2 v + \alpha_3)}\frac{dW}{dt}. \tag{63}$$

Hence, the stationary distribution becomes

$$P(v) \propto \frac{v^{\beta_2/2\alpha_3 T - 3/2 - \beta_2^2/2\alpha_3}}{(\alpha_1 v^2 - 2\alpha_2 v + \alpha_3)^{1+\beta_2/4T\alpha_3 + K_1}} \exp\left(-\left(\frac{1}{2T}\frac{\alpha_3\beta_1 - \alpha_2\beta_2}{\alpha_3\sqrt{\Delta}} + K_2\right)\arctan\frac{\alpha_1 v - \alpha_2}{\sqrt{\Delta}}\right), \tag{64}$$

where

$$K_1 = \frac{3\alpha_3\beta_1^2 - \alpha_1\beta_2^2}{4\alpha_1\alpha_3},$$

$$K_2 = \frac{3\alpha_2\alpha_3\beta_1^2 - 4\alpha_1\alpha_3\beta_1\beta_2 + \alpha_1\alpha_2\beta_2^2}{2\alpha_1\alpha_3\sqrt{\Delta}}. \tag{65}$$

From the expression above we can see $K_1 \ll 1 + \beta_2/4T\alpha_3$ and $K_2 \ll (\alpha_3\beta_1 - \alpha_2\beta_2)/2T\alpha_3\sqrt{\Delta}$. Hence, the effect of modification can only be seen in the term proportional to $v$. The phase transition point is modified as

$$T_c = \frac{\beta_2}{\alpha_3 + \beta_2^2}. \tag{66}$$

Compared with the previous result $T_c = \frac{\beta_2}{\alpha_3}$, we can see the effect of the loss modification is $\alpha_3 \rightarrow \alpha_3 + \beta_2^2$, or equivalently, $\text{Var}[xy] \rightarrow \mathbb{E}[x^2 y^2]$. This effect can be seen from $E_1$ and $E_2$.

## A.4   PROOF OF THEOREM 3.3

*Proof.* For any $i$, one can obtain the expressions of $C_1^{(i)}$ and $C_2^{(i)}$ from Theorem 3.1 as

$$(C_1^{(i)})_{\alpha_1,\alpha_2} = 4p_i \mathbb{E}_i \left[ \|\tilde{x}\|^2 (\sum_{j=1}^d u_j^{\alpha_1} v_j^T \tilde{x} - y^{\alpha_1})(\sum_{j=1}^d u_j^{\alpha_2} v_j^T \tilde{x} - y^{\alpha_2}) \right]$$

$$- 4p_i^2 \sum_\beta \mathbb{E}_i \left[ \tilde{x}^\beta (\sum_{j=1}^d u_j^{\alpha_1} v_j^T \tilde{x} - y^{\alpha_1}) \right] \mathbb{E}_i \left[ \tilde{x}^\beta (\sum_{j=1}^d u_j^{\alpha_2} v_j^T \tilde{x} - y^{\alpha_2}) \right]$$

$$= 4p_i \mathbb{E}_i \left[ \|\tilde{x}\|^2 r^{\alpha_1} r^{\alpha_2} \right] - 4p_i^2 \sum_\beta \mathbb{E}_i \left[ \tilde{x}^\beta r^{\alpha_1} \right] \mathbb{E}_i \left[ \tilde{x}^\beta r^{\alpha_2} \right], \tag{67}$$

$$(C_2^{(i)})_{\beta_1,\beta_2} = 4\mathbb{E}_i \left[ \tilde{x}^{\beta_1} \tilde{x}^{\beta_2} \| \sum_{j=1}^d u_j v_j^T \tilde{x} - y \|^2 \right] - 4\sum_\alpha \mathbb{E}_i \left[ \tilde{x}^{\beta_1} (\sum_{j=1}^d u_j^\alpha v_j^T \tilde{x} - y^\alpha) \right] \mathbb{E}_i \left[ \tilde{x}^{\beta_2} (\sum_{j=1}^d u_j^\alpha v_j^T \tilde{x} - y^\alpha) \right]$$

$$= 4p_i \mathbb{E}_i \left[ \|r\|^2 \tilde{x}^{\beta_1} \tilde{x}^{\beta_2} \right] - 4p_i^2 \sum_\alpha \mathbb{E}_i \left[ \tilde{x}^{\beta_1} r^\alpha \right] \mathbb{E}_i \left[ \tilde{x}^{\beta_2} r^\alpha \right], \tag{68}$$

where we use the notation $r^\alpha := \sum_{j=1}^d u_j^\alpha v_j^T \tilde{x} - y^\alpha$, $\tilde{x} := (x^T, 1)^T$, $v_i = (w_i^T, b_i)^T$ and $\mathbb{E}_i[O] := \mathbb{E}[O | w_i^T x + b_i > 0]$.

We start with showing that $C_1^{(1)}$ is full-rank. Let $m$ be an arbitrary unit vector in $\mathbb{R}^{d_u}$. We have that

$$m^T C_1^{(i)} m = 4p_i \mathbb{E}_i \left[ \|\tilde{x}\|^2 (m^T r)^2 \right] - 4p_i^2 \sum_\beta \mathbb{E}_i \left[ \tilde{x}^\beta (m^T r) \right] \mathbb{E}_i \left[ \tilde{x}^\beta (m^T r) \right]$$

$$\geq 4p_i^2 \mathbb{E}_i \left[ \|\tilde{x}\|^2 (m^T r)^2 \right] - 4p_i^2 \sum_\beta \mathbb{E}_i \left[ \tilde{x}^\beta (m^T r) \right] \mathbb{E}_i \left[ \tilde{x}^\beta (m^T r) \right]$$

$$= 4p_i^2 \sum_\beta \mathrm{Var}_i [\tilde{x}^\beta m^T r]$$

$$= 4p_i^2 \sum_\beta [\mathrm{Var}_i [\tilde{x}^\beta m^T (g(x) - \sum_{j=1}^d u_j v_j^T \tilde{x})] + \mathrm{Var}_i [\tilde{x}^\beta m^T \epsilon] - 2\mathrm{Cov}_i [\tilde{x}^\beta m^T (g(x) - \sum_{j=1}^d u_j v_j^T \tilde{x}), \tilde{x}^\beta m^T \epsilon]]$$

$$\geq 4p_i^2 \sum_\beta \mathrm{Var}_i [\tilde{x}^\beta m^T \epsilon] > 0, \tag{69}$$

where the last inequality follows from

$$\mathrm{Cov}[\tilde{x}^\beta m^T (g(x) - \sum_{j=1}^d u_j v_j^T \tilde{x}), \tilde{x}^\beta m^T \epsilon]$$

$$= \mathbb{E}_i [(\tilde{x}^\beta)^2 m^T (g(x) - \sum_{j=1}^d u_j v_j^T \tilde{x}) m^T \epsilon] - \mathbb{E}_i [\tilde{x}^\beta m^T (g(x) - \sum_{j=1}^d u_j v_j^T \tilde{x})] \mathbb{E}_i [\tilde{x}^\beta m^T \epsilon]$$

$$= 0. \tag{70}$$

Here we denote that $\mathrm{Var}_i[O] := \mathbb{E}_i[O^2] - \mathbb{E}_i[O]^2$ and $\mathrm{Cov}_i[O_1, O_2] := \mathbb{E}_i[O_1 O_2] - \mathbb{E}_i[O_1]\mathbb{E}_i[O_2]$.

For $C_2^{(i)}$, we let the vector $\tilde{n} := (n^T, n_f)^T$ be a unit vector in $\mathbb{R}^{d_w+1}$, yielding

$$\tilde{n}^T C_2^{(i)} \tilde{n} = 4p_i \mathbb{E}_i \left[ \|r\|^2 (\tilde{n}^T \tilde{x})^2 \right] - 4p_i^2 \sum_\alpha \mathbb{E}_i \left[ r^\alpha (\tilde{n}^T \tilde{x}) \right] \mathbb{E}_i \left[ r^\alpha (\tilde{n}^T \tilde{x}) \right]$$

$$\geq 4p_i^2 \mathbb{E}_i \left[ \|r\|^2 (\tilde{n}^T \tilde{x})^2 \right] - 4p_i^2 \sum_\alpha \mathbb{E}_i \left[ r^\alpha (\tilde{n}^T \tilde{x}) \right] \mathbb{E}_i \left[ r^\alpha (\tilde{n}^T \tilde{x}) \right]$$

$$= 4p_i^2 \sum_\alpha \mathrm{Var}_i [r^\alpha \tilde{n}^T \tilde{x}]. \tag{71}$$

Note that this quantity can be decomposed as

$$\sum_\alpha \mathrm{Var}_i[r^\alpha \tilde{n}^T \tilde{x}] = \sum_\alpha \mathrm{Var}_i\Big[\big(g^\alpha(x) - \sum_{j=1}^d u_j^\alpha v_j^T \tilde{x} + \epsilon^\alpha\big)(\tilde{n}^T \tilde{x})\Big]$$

$$= \sum_\alpha \mathrm{Var}_i\Big[\big(g^\alpha(x) - \sum_{j=1}^d u_j^\alpha v_j^T \tilde{x}\big)(n^T x + n_f)\Big] + \sum_\alpha \mathrm{Var}_i[\epsilon^\alpha(n^T x + n_f)]$$

$$- 2\sum_\alpha \mathrm{Cov}_i\Big[\big(g^\alpha(x) - \sum_{j=1}^d u_j^\alpha v_j^T \tilde{x}\big)(n^T x + n_f), \epsilon^\alpha(n^T x + n_f)\Big]. \tag{72}$$

The covariance term vanishes because

$$\mathrm{Cov}\Big[\big(g^\alpha(x) - \sum_{j=1}^d u_j^\alpha v_j^T \tilde{x}\big)(n^T x + n_f), \epsilon^\alpha(n^T x + n_f)\Big]$$

$$= \mathbb{E}_i\Big[\big(g^\alpha(x) - \sum_{j=1}^d u_j^\alpha v_j^T \tilde{x}\big)\epsilon^\alpha(n^T x + n_f)^2\Big] - \mathbb{E}_i\Big[\big(g^\alpha(x) - \sum_{j=1}^d u_j^\alpha v_j^T \tilde{x}\big)(n^T x + n_f)\Big]\mathbb{E}_i[\epsilon^\alpha(n^T x + n_f)]$$

$$= 0. \tag{73}$$

Therefore,

$$\tilde{n}^T C_2^{(i)} \tilde{n} \geq \sum_\alpha \mathrm{Var}_i\Big[\big(g^\alpha(x) - \sum_{j=1}^d u_j^\alpha v_j^T \tilde{x}\big)(n^T x + n_f)\Big] + \sum_\alpha \mathrm{Var}_i[\epsilon^\alpha(n^T x + n_f)]$$

$$\geq \sum_\alpha \mathrm{Var}_i[\epsilon^\alpha(n^T x + n_f)]$$

$$= \sum_\alpha \mathrm{Var}_i[\epsilon^\alpha]\mathrm{Var}_i[(n^T x + n_f)] + \sum_\alpha \big(\mathrm{Var}_i[\epsilon^\alpha]\mathbb{E}_i[(n^T x + n_f)^2] + \mathrm{Var}_i[n^T x + n_f]\mathbb{E}_i[(\epsilon^\alpha)^2]\big)$$

$$\geq \sum_\alpha \mathrm{Var}_i[\epsilon^\alpha]\mathbb{E}_i[(n^T x + n_f)^2] > 0, \tag{74}$$

where the penultimate inequality follows from the fact that $\epsilon$ is independent of $x$. Hence, both the matrices $C_1^{(i)}$ and $C_2^{(i)}$ are full-rank. The proof is completed. $\qquad\square$

## A.5 DERIVATION OF EQ. (6)

We here prove inequality (6). At stationarity, $d(\|u\|^2 - \|w\|^2)/dt = 0$, indicating

$$\lambda_{1M}\|u\|^2 - \lambda_{2m}\|w\|^2 \geq 0, \ \lambda_{1m}\|u\|^2 - \lambda_{2M}\|w\|^2 \leq 0. \tag{75}$$

The first inequality in Eq. (75) gives the solution

$$\frac{\|u\|^2}{\|w\|^2} \geq \frac{\lambda_{2m}}{\lambda_{1M}}. \tag{76}$$

The second inequality in Eq. (75) gives the solution

$$\frac{\|u\|^2}{\|w\|^2} \leq \frac{\lambda_{2M}}{\lambda_{1m}}. \tag{77}$$

Combining these two results, we obtain

$$\frac{\lambda_{2m}}{\lambda_{1M}} \leq \frac{\|u\|^2}{\|w\|^2} \leq \frac{\lambda_{2M}}{\lambda_{1m}}, \tag{78}$$

which is Eq. (6).

## A.6 PROOF OF THEOREM 4.1

*Proof.* This proof is based on the fact that if a certain condition is satisfied for all trajectories with probability 1, this condition is satisfied by the stationary distribution of the dynamics with probability 1.

Let us first consider the case of $D > 1$. We first show that any trajectory satisfies at least one of the following five conditions: for any $i$, (i) $v_i \to 0$, (ii) $L(\theta) \to 0$, or (iii) for any $k \neq l$, $(u_i^{(k)})^2 - (u_i^{(l)})^2 \to 0$.

The SDE for $u_i^{(k)}$ is

$$\frac{du_i^{(k)}}{dt} = -2\frac{v_i}{u_i^{(k)}}(\beta_1 v - \beta_2) + 2\frac{v_i}{u_i^{(k)}}\sqrt{\eta(\alpha_1 v^2 - 2\alpha_2 v + \alpha_3)}\frac{dW}{dt}, \tag{79}$$

where $v_i := \prod_{k=1}^{D} u_i^{(k)}$, and so $v = \sum_i v_i$. There exists rescaling symmetry between $u_i^{(k)}$ and $u_i^{(l)}$ for $k \neq l$. By the law of balance, we have

$$\frac{d}{dt}[(u_i^{(k)})^2 - (u_i^{(l)})^2] = -T[(u_i^{(k)})^2 - (u_i^{(l)})^2]\text{Var}\left[\frac{\partial\ell}{\partial(u_i^{(k)}u_i^{(l)})}\right], \tag{80}$$

where

$$\text{Var}\left[\frac{\partial\ell}{\partial(u_i^{(k)}u_i^{(l)})}\right] = \left(\frac{v_i}{u_i^{(k)}u_i^{(l)}}\right)^2(\alpha_1 v^2 - 2\alpha_2 v + \alpha_3) \tag{81}$$

with $v_i/(u_i^{(k)}u_i^{(l)}) = \prod_{s\neq k,l} u_i^{(s)}$. In the long-time limit, $(u_i^{(k)})^2$ converges to $(u_i^{(l)})^2$ unless $\text{Var}\left[\frac{\partial\ell}{\partial(u_i^{(k)}u_i^{(l)})}\right] = 0$, which is equivalent to $v_i/(u_i^{(k)}u_i^{(l)}) = 0$ or $\alpha_1 v^2 - 2\alpha_2 v + \alpha_3 = 0$. These two conditions correspond to conditions (i) and (ii). The latter is because $\alpha_1 v^2 - 2\alpha_2 v + \alpha_3 = 0$ takes place if and only if $v = \alpha_2/\alpha_1$ and $\alpha_2^2 - \alpha_1\alpha_3 = 0$ together with $L(\theta) = 0$. Therefore, at stationarity, we must have conditions (i), (ii), or (iii).

Now, we prove that when (iii) holds, the condition 2-(b) in the theorem statement must hold: for $D = 1$, $(\log|v_i| - \log|v_j|) = c_0$ with $\text{sgn}(v_i) = \text{sgn}(v_j)$. When (iii) holds, there are two situations. First, if $v_i = 0$, we have $u_i^{(k)} = 0$ for all $k$, and $v_i$ will stay 0 for the rest of the trajectory, which corresponds to condition (i).

If $v_i \neq 0$, we have $u_i^{(k)} \neq 0$ for all $k$. Therefore, the dynamics of $v_i$ is

$$\frac{dv_i}{dt} = -2\sum_k \left(\frac{v_i}{u_i^{(k)}}\right)^2(\beta_1 v - \beta_2) + 2\sum_k \left(\frac{v_i}{u_i^{(k)}}\right)^2\sqrt{\eta(\alpha_1 v^2 - 2\alpha_2 v + \alpha_3)}\frac{dW}{dt} + 4\sum_{k,l}\left(\frac{v_i^3}{(u_i^{(k)}u_i^{(l)})^2}\right)\eta(\alpha_1 v^2 - 2\alpha_2 v + \alpha_3). \tag{82}$$

Comparing the dynamics of $v_i$ and $v_j$ for $i \neq j$, we obtain

$$\frac{dv_i/dt}{\sum_k(v_i/u_i^{(k)})^2} - \frac{dv_j/dt}{\sum_k(v_j/u_j^{(k)})^2} = 4\left(\frac{\sum_{m,l} v_i^3/(u_i^{(m)}u_i^{(l)})^2}{\sum_k(v_i/u_i^{(k)})^2} - \frac{\sum_{m,l} v_j^3/(u_j^{(m)}u_j^{(l)})^2}{\sum_k(v_j/u_j^{(k)})^2}\right)\eta(\alpha_1 v^2 - 2\alpha_2 v + \alpha_3)$$

$$= 4\left(v_i\frac{\sum_{m,l} v_i^2/(u_i^{(m)}u_i^{(l)})^2}{\sum_k(v_i/u_i^{(k)})^2} - v_j\frac{\sum_{m,l} v_j^2/(u_j^{(m)}u_j^{(l)})^2}{\sum_k(v_j/u_j^{(k)})^2}\right)\eta(\alpha_1 v^2 - 2\alpha_2 v + \alpha_3). \tag{83}$$

By condition (iii), we have $|u_i^{(0)}| = \cdots = |u_i^{(D)}|$, i.e., $(v_i/u_i^{(k)})^2 = (v_i^2)^{D/(D+1)}$ and $(v_i/u_i^{(m)}u_i^{(l)})^2 = (v_i^2)^{(D-1)/(D+1)}$.[6] Therefore, we obtain

$$\frac{dv_i/dt}{(D+1)(v_i^2)^{D/(D+1)}} - \frac{dv_j/dt}{(D+1)(v_j^2)^{D/(D+1)}} = \left(v_i\frac{D(v_i^2)^{(D-1)/(D+1)}}{2(v_i^2)^{D/(D+1)}} - v_j\frac{D(v_j^2)^{(D-1)/(D+1)}}{2(v_j^2)^{D/(D+1)}}\right)\eta(\alpha_1 v^2 - 2\alpha_2 v + \alpha_3). \tag{84}$$

We first consider the case where $v_i$ and $v_j$ initially share the same sign (both positive or both negative). When $D > 1$, the left-hand side of Eq. (84) can be written as

$$\frac{1}{1-D}\frac{dv_i^{2/(D+1)-1}}{dt} + 4Dv_i^{1-2/(D+1)}\eta(\alpha_1 v^2 - 2\alpha_2 v + \alpha_3) - \frac{1}{1-D}\frac{dv_j^{2/(D+1)-1}}{dt} - 4Dv_j^{1-2/(D+1)}\eta(\alpha_1 v^2 - 2\alpha_2 v + \alpha_3), \tag{85}$$

---

[6]Here, we only consider the root on the positive real axis.

which follows from Ito's lemma:

$$\frac{dv_i^{2/(D+1)-1}}{dt} = \left(\frac{2}{D+1}-1\right)v_i^{2/(D+1)-2}\frac{dv_i}{dt} + 2(\frac{2}{D+1}-1)(\frac{2}{D+1}-2)v_i^{2/(D+1)-3}\left(\sum_k(\frac{v_i}{u_i^{(k)}})^2\sqrt{\eta(\alpha_1 v^2 - 2\alpha_2 v + \alpha_3)}\right)^2$$

$$= (\frac{2}{D+1}-1)v_i^{2/(D+1)-2}\frac{dv_i}{dt} + 4D(D-1)v_i^{1-2/(D+1)}\eta(\alpha_1 v^2 - 2\alpha_2 v + \alpha_3). \qquad (86)$$

Substitute in Eq. (84), we obtain Eq. (85).

Now, we consider the right-hand side of Eq. (84), which is given by

$$2Dv_i^{1-2/(D+1)}\eta(\alpha_1 v^2 - 2\alpha_2 v + \alpha_3) - 2Dv_j^{1-2/(D+1)}\eta(\alpha_1 v^2 - 2\alpha_2 v + \alpha_3). \qquad (87)$$

Combining Eq. (85) and Eq. (87), we obtain

$$\frac{1}{1-D}\frac{dv_i^{2/(D+1)-1}}{dt} - \frac{1}{1-D}\frac{dv_j^{2/(D+1)-1}}{dt} = -2D(v_i^{1-2/(D+1)} - v_j^{1-2/(D+1)})\eta(\alpha_1 v^2 - 2\alpha_2 v + \alpha_3). \qquad (88)$$

By defining $z_i = v_i^{2/(D+1)-1}$, we can further simplify the dynamics:

$$\frac{d(z_i - z_j)}{dt} = 2D(D-1)\left(\frac{1}{z_i} - \frac{1}{z_j}\right)\eta(\alpha_1 v^2 - 2\alpha_2 v + \alpha_3)$$

$$= -2D(D-1)\frac{z_i - z_j}{z_i z_j}\eta(\alpha_1 v^2 - 2\alpha_2 v + \alpha_3). \qquad (89)$$

Hence,

$$z_i(t) - z_j(t) = \exp\left[-\int dt \frac{2D(D-1)}{z_i z_j}\eta(\alpha_1 v^2 - 2\alpha_2 v + \alpha_3)\right]. \qquad (90)$$

Therefore, if $v_i$ and $v_j$ initially have the same sign, they will decay to the same value in the long-time limit $t \to \infty$, which gives condition 2-(b). When $v_i$ and $v_j$ initially have different signs, we can write Eq. (84) as

$$\frac{d|v_i|/dt}{(D+1)(|v_i|^2)^{D/(D+1)}} + \frac{d|v_j|/dt}{(D+1)(|v_j|^2)^{D/(D+1)}} = \left(|v_i|\frac{D(|v_i|^2)^{(D-1)/(D+1)}}{2(|v_i|^2)^{D/(D+1)}} + |v_j|\frac{D(|v_j|^2)^{(D-1)/(D+1)}}{2(|v_j|^2)^{D/(D+1)}}\right)$$

$$\times \eta(\alpha_1 v^2 - 2\alpha_2 v + \alpha_3). \qquad (91)$$

Hence, when $D > 1$, we simplify the equation with a similar procedure as

$$\frac{1}{1-D}\frac{d|v_i|^{2/(D+1)-1}}{dt} + \frac{1}{1-D}\frac{d|v_j|^{2/(D+1)-1}}{dt} = -2D(|v_i|^{1-2/(D+1)} + |v_j|^{1-2/(D+1)})\eta(\alpha_1 v^2 - 2\alpha_2 v + \alpha_3). \qquad (92)$$

Defining $z_i = |v_i|^{2/(D+1)-1}$, we obtain

$$\frac{d(z_i + z_j)}{dt} = 2D(D-1)\left(\frac{1}{z_i} + \frac{1}{z_j}\right)\eta(\alpha_1 v^2 - 2\alpha_2 v + \alpha_3)$$

$$= 2D(D-1)\frac{z_i + z_j}{z_i z_j}\eta(\alpha_1 v^2 - 2\alpha_2 v + \alpha_3), \qquad (93)$$

which implies

$$z_i(t) + z_j(t) = \exp\left[\int dt \frac{2D(D-1)}{z_i z_j}\eta(\alpha_1 v^2 - 2\alpha_2 v + \alpha_3)\right]. \qquad (94)$$

From this equation, we reach the conclusion that if $v_i$ and $v_j$ have different signs initially, one of them converges to 0 in the long-time limit $t \to \infty$, corresponding to condition 1 in the theorem statement. Hence, for $D > 1$, at least one of the conditions is always satisfied at $t \to \infty$.

Now, we prove the theorem for $D = 1$, which is similar to the proof above. The law of balance gives

$$\frac{d}{dt}[(u_i^{(1)})^2 - (u_i^{(2)})^2] = -T[(u_i^{(1)})^2 - (u_i^{(2)})^2]\text{Var}\left[\frac{\partial \ell}{\partial(u_i^{(1)}u_i^{(2)})}\right]. \qquad (95)$$

We can see that $|u_i^{(1)}| \to |u_i^{(2)}|$ takes place unless $\mathrm{Var}\left[\frac{\partial \ell}{\partial(u_i^{(1)} u_i^{(2)})}\right] = 0$, which is equivalent to $L(\theta) = 0$. This corresponds to condition (ii). Hence, if condition (ii) is violated, we need to prove condition (iii). In this sense, $|u_i^{(1)}| \to |u_i^{(2)}|$ occurs and Eq. (84) can be rewritten as

$$\frac{dv_i/dt}{|v_i|} - \frac{dv_j/dt}{|v_j|} = (\mathrm{sign}(v_i) - \mathrm{sign}(v_j))\eta(\alpha_1 v^2 - 2\alpha_2 v + \alpha_3). \tag{96}$$

When $v_i$ and $v_j$ are both positive, we have

$$\frac{dv_i/dt}{v_i} - \frac{dv_j/dt}{v_j} = 0. \tag{97}$$

With Ito's lemma, we have

$$\frac{d\log(v_i)}{dt} = \frac{dv_i}{v_i dt} - 2\eta(\alpha_1 v^2 - 2\alpha_2 v + \alpha_3). \tag{98}$$

Therefore, Eq. (97) can be simplified to

$$\frac{d(\log(v_i) - \log(v_j))}{dt} = 0, \tag{99}$$

which indicates that all $v_i$ with the same sign will decay at the same rate. This differs from the case of $D > 2$ where all $v_i$ decay to the same value. Similarly, we can prove the case where $v_i$ and $v_j$ are both negative.

Now, we consider the case where $v_i$ is positive while $v_j$ is negative and rewrite Eq. (96) as

$$\frac{dv_i/dt}{v_i} + \frac{d(|v_j|)/dt}{|v_j|} = 2\eta(\alpha_1 v^2 - 2\alpha_2 v + \alpha_3). \tag{100}$$

Furthermore, we can derive the dynamics of $v_j$ with Ito's lemma:

$$\frac{d\log(|v_j|)}{dt} = \frac{dv_i}{v_i dt} - 2\eta(\alpha_1 v^2 - 2\alpha_2 v + \alpha_3). \tag{101}$$

Therefore, Eq. (100) takes the form of

$$\frac{d(\log(v_i) + \log(|v_j|))}{dt} = -2\eta(\alpha_1 v^2 - 2\alpha_2 v + \alpha_3). \tag{102}$$

In the long-time limit, we can see $\log(v_i|v_j|)$ decays to $-\infty$, indicating that either $v_i$ or $v_j$ will decay to 0. This corresponds to condition 1 in the theorem statement. Combining Eq. (99) and Eq. (102), we conclude that all $v_i$ have the same sign as $t \to \infty$, which indicates condition 2-(a) if conditions in item 1 are all violated. The proof is thus complete. $\qquad\square$

## A.7 Proof of Theorem 4.2

*Proof.* Following Eq. (82), we substitute $u_i^{(k)}$ with $v_i^{1/D}$ for arbitrary $k$ and obtain

$$\frac{dv_i}{dt} = -2(D+1)|v_i|^{2D/(D+1)}(\beta_1 v - \beta_2) + 2(D+1)|v_i|^{2D/(D+1)}\sqrt{\eta(\alpha_1 v^2 - 2\alpha_2 v + \alpha_3)}\frac{dW}{dt}$$
$$+ 2(D+1)Dv_i^3|v_i|^{-4/(D+1)}\eta(\alpha_1 v^2 - 2\alpha_2 v + \alpha_3). \tag{103}$$

With Eq. (90), we can see that for arbitrary $i$ and $j$, $v_i$ will converge to $v_j$ in the long-time limit. In this case, we have $v = dv_i$ for each $i$. Then, the SDE for $v$ can be written as

$$\frac{dv}{dt} = -2(D+1)d^{2/(D+1)-1}|v|^{2D/(D+1)}(\beta_1 v - \beta_2) + 2(D+1)d^{2/(D+1)-1}|v|^{2D/(D+1)}\sqrt{\eta(\alpha_1 v^2 - 2\alpha_2 v + \alpha_3)}\frac{dW}{dt}$$
$$+ 2(D+1)Dd^{4/(D+1)-2}v^3|v|^{-4/(D+1)}\eta(\alpha_1 v^2 - 2\alpha_2 v + \alpha_3). \tag{104}$$

If $v > 0$, Eq. (104) becomes

$$\frac{dv}{dt} = -2(D+1)d^{2/(D+1)-1}v^{2D/(D+1)}(\beta_1 v - \beta_2) + 2(D+1)d^{2/(D+1)-1}v^{2D/(D+1)}\sqrt{\eta(\alpha_1 v^2 - 2\alpha_2 v + \alpha_3)}\frac{dW}{dt}$$
$$+ 2(D+1)Dd^{4/(D+1)-2}v^{3-4/(D+1)}\eta(\alpha_1 v^2 - 2\alpha_2 v + \alpha_3). \tag{105}$$

Therefore, the stationary distribution of a general deep diagonal network is given by

$$p(v) \propto \frac{1}{v^{3(1-1/(D+1))}(\alpha_1 v^2 - 2\alpha_2 v + \alpha_3)} \exp\left(-\frac{1}{T}\int dv \frac{d^{1-2/(D+1)}(\beta_1 v - \beta_2)}{(D+1)v^{2D/(D+1)}(\alpha_1 v^2 - 2\alpha_2 v + \alpha_3)}\right). \tag{106}$$

If $v < 0$, Eq. (104) becomes

$$\frac{d|v|}{dt} = -2(D+1)d^{2/(D+1)-1}|v|^{2D/(D+1)}(\beta_1|v| + \beta_2) - 2(D+1)d^{2/(D+1)-1}|v|^{2D/(D+1)}\sqrt{\eta(\alpha_1|v|^2 + 2\alpha_2|v| + \alpha_3)}\frac{dW}{dt}$$
$$+ 2(D+1)Dd^{4/(D+1)-2}|v|^{3-4/(D+1)}\eta(\alpha_1|v|^2 + 2\alpha_2|v| + \alpha_3). \tag{107}$$

The stationary distribution of $|v|$ is given by

$$p(|v|) \propto \frac{1}{|v|^{3(1-1/(D+1))}(\alpha_1|v|^2 + 2\alpha_2|v| + \alpha_3)} \exp\left(-\frac{1}{T}\int d|v| \frac{d^{1-2/(D+1)}(\beta_1|v| + \beta_2)}{(D+1)|v|^{2D/(D+1)}(\alpha_1|v|^2 + 2\alpha_2|v| + \alpha_3)}\right). \tag{108}$$

Thus, we have obtained

$$p_\pm(|v|) \propto \frac{1}{|v|^{3(1-1/(D+1))}(\alpha_1|v|^2 \mp 2\alpha_2|v| + \alpha_3)} \exp\left(-\frac{1}{T}\int d|v| \frac{d^{1-2/(D+1)}(\beta_1|v| \mp \beta_2)}{(D+1)|v|^{2D/(D+1)}(\alpha_1|v|^2 \mp 2\alpha_2|v| + \alpha_3)}\right). \tag{109}$$

Especially when $D = 1$, the distribution function can be simplified as

$$p_\pm(|v|) \propto \frac{|v|^{\pm\beta_2/2\alpha_3 T - 3/2}}{(\alpha_1|v|^2 \mp 2\alpha_2|v| + \alpha_3)^{1\pm\beta_2/4T\alpha_3}} \exp\left(-\frac{1}{2T}\frac{\alpha_3\beta_1 - \alpha_2\beta_2}{\alpha_3\sqrt{\Delta}} \arctan\frac{\alpha_1|v| \mp \alpha_2}{\sqrt{\Delta}}\right), \quad (110)$$

where we have used the integral

$$\int dv \frac{\beta_1 v \mp \beta_2}{v(\alpha_1 v^2 - 2\alpha_2 v + \alpha_3)} = \frac{\alpha_3\beta_1 - \alpha_2\beta_2}{\alpha_3\sqrt{\Delta}} \arctan\frac{\alpha_1|v| \mp \alpha_2}{\sqrt{\Delta}} \pm \frac{\beta_2}{\alpha_3}\log(v) \pm \frac{\beta_2}{2\alpha_3}\log(\alpha_1 v^2 - 2\alpha_2 v + \alpha_3). \tag{111}$$

Furthermore, we can also see that $p(v) = \delta(v)$ is also the stationary distribution of the Fokker-Planck equation of Eq. (105). Hence, the general stationary distribution of $v$ can be expressed as

$$p^*(v) = (1 - z)\delta(v) + zp_\pm(v). \tag{112}$$

The proof is complete. $\qquad\square$

A.8 ANALYSIS OF THE MAXIMUM PROBABILITY POINT

To investigate the existence of the maximum point given in Eq. (18), we treat $T$ as a variable and study whether $(\beta_1 - 10\alpha_2 T)^2 + 28\alpha_1 T(\beta_2 - 3\alpha_3 T) := A$ in the square root is always positive or not. When $T < \frac{\beta_2}{3\alpha_3} = T_c/3$, $A$ is positive for arbitrary data. When $T > \frac{\beta_2}{3\alpha_3}$, we divide the discussion into several cases. First, when $\alpha_1\alpha_3 > \frac{25}{21}\alpha_2^2$, there exists a root for the expression $A$. Hence, we find that

$$T = \frac{-5\alpha_2\beta_1 + 7\alpha_1\beta_2 + \sqrt{7}\sqrt{3\alpha_1\alpha_3\beta_1^2 - 10\alpha_1\alpha_2\beta_1\beta_2 + 7\alpha_1^2\beta_2^2}}{2(21\alpha_1\alpha_3 - 25\alpha_2^2)} := T^* \tag{113}$$

is a critical point. When $T_c/3 < T < T^*$, there exists a solution to the maximum condition. When $T > T^*$, there is no solution to the maximum condition.

The second case is $\alpha_2^2 < \alpha_1\alpha_3 < \frac{25}{21}\alpha_2^2$. In this case, we need to further compare the value between $5\alpha_2\beta_1$ and $7\alpha_1\beta_2$. If $5\alpha_2\beta_1 < 7\alpha_1\beta_2$, we have $A > 0$, which indicates that the maximum point exists. If $5\alpha_2\beta_1 > 7\alpha_1\beta_2$, we need to further check the value of minimum of $A$, which takes the form of

$$\min_T A(T) = \frac{(25\alpha_2^2 - 21\alpha_1\alpha_3)\beta_1^2 - (7\alpha_1\beta_2 - 5\alpha_2\beta_1)^2}{25\alpha_2^2 - 21\alpha_1\alpha_3}. \tag{114}$$

If $\frac{7\alpha_1}{5\alpha_2} < \frac{\beta_1}{\beta_2} < \frac{5\alpha_2 + \sqrt{25\alpha_2^2 - 21\alpha_1\alpha_3}}{3\alpha_3}$, the minimum of $A$ is positive and the maximum exists. However, if $\frac{\beta_1}{\beta_2} \geq \frac{5\alpha_2 + \sqrt{25\alpha_2^2 - 21\alpha_1\alpha_3}}{3\alpha_3}$, there is a critical learning rate $T^*$. If $\frac{\beta_1}{\beta_2} = \frac{5\alpha_2 + \sqrt{25\alpha_2^2 - 21\alpha_1\alpha_3}}{3\alpha_3}$, there is

| | without weight decay | with weight decay |
|---|---|---|
| single layer | $(\alpha_1 v^2 - 2\alpha_2 v + \alpha_3)^{-1-\frac{\beta_1}{2T\alpha_1}}$ | $\alpha_1(v-k)^{-2-\frac{(\beta_1+\gamma)}{T\alpha_1}}$ |
| non-interpolation | $\dfrac{v^{\beta_2/2\alpha_3 T - 3/2}}{(\alpha_1 v^2 - 2\alpha_2 v + \alpha_3)^{1+\beta_2/4T\alpha_3}}$ | $\dfrac{v^{S(\beta_2-\gamma)/2\alpha_3\lambda - 3/2}}{(\alpha_1 v^2 - 2\alpha_2 v + \alpha_3)^{1+(\beta_2-\gamma)/4T\alpha_3}}$ |
| interpolation $y = kx$ | $\dfrac{v^{-3/2+\beta_1/2T\alpha_1 k}}{(v-k)^{2+\beta_1/2T\alpha_1 k}}$ | $\dfrac{v^{-3/2+\frac{1}{2T\alpha_1 k}(\beta_1-\frac{\gamma}{k})}}{(v-k)^{2+\frac{1}{2T\alpha_1 k}(\beta_1-\frac{\gamma}{k})}} \exp\left(-\frac{\beta\gamma}{2T\alpha_1}\frac{1}{k(k-v)}\right)$ |

Table 1: Summary of distributions $p(v)$ in a depth-1 neural network. Here, we show the distribution in the nontrivial subspace when the data $x$ and $y$ are positively correlated. The $\Theta(1)$ factors are neglected for concision.

only one critical learning rate as $T_c = \frac{5\alpha_2\beta_1 - 7\alpha_1\beta_2}{2(25\alpha_2^2 - 21\alpha_1\alpha_3)}$. When $T_c/3 < T < T^*$, there is a solution to the maximum condition, while there is no solution when $T > T^*$. If $\frac{\beta_1}{\beta_2} > \frac{5\alpha_2 + \sqrt{25\alpha_2^2 - 21\alpha_1\alpha_3}}{3\alpha_3}$, there are two critical points:

$$T_{1,2} = \frac{-5\alpha_2\beta_1 + 7\alpha_1\beta_2 \mp \sqrt{7}\sqrt{3\alpha_1\alpha_3\beta_1^2 - 10\alpha_1\alpha_2\beta_1\beta_2 + 7\alpha_1^2\beta_2^2}}{2(21\alpha_1\alpha_3 - 25\alpha_2^2)}. \tag{115}$$

For $T < T_1$ and $T > T_2$, there exists a solution to the maximum condition. For $T_1 < T < T_2$, there is no solution to the maximum condition. The last case is $\alpha_2^2 = \alpha_1\alpha_3 < \frac{25}{21}\alpha_2^2$. In this sense, the expression of $A$ is simplified as $\beta_1^2 + 28\alpha_1\beta_2 T - 20\alpha_2\beta_1 T$. Hence, when $\frac{\beta_1}{\beta_2} < \frac{7\alpha_1}{5\alpha_2}$, there is no critical learning rate and the maximum always exists. Nonetheless, when $\frac{\beta_1}{\beta_2} > \frac{7\alpha_1}{5\alpha_2}$, there is a critical learning rate as $T^* = \frac{\beta_1^2}{20\alpha_2\beta_1 - 28\alpha_1\beta_2}$. When $T < T^*$, there is a solution to the maximum condition, while there is no solution when $T > T^*$.

A.9 Other Cases for $D = 1$

The other cases are worth studying. For the interpolation case where the data is linear ($y = kx$ for some $k$), the stationary distribution is different and simpler. There exists a nontrivial fixed point for $\sum_i(u_i^2 - w_i^2)$: $\sum_j u_j w_j = \frac{\alpha_2}{\alpha_1}$, which is the global minimizer of $L$ and also has a vanishing noise. It is helpful to note the following relationships for the data distribution when it is linear:

$$\begin{cases} \alpha_1 = \mathrm{Var}[x^2], \\ \alpha_2 = k\mathrm{Var}[x^2] = k\alpha_1, \\ \alpha_3 = k^2\alpha_1, \\ \beta_1 = \mathbb{E}[x^2], \\ \beta_2 = k\mathbb{E}[x^2] = k\beta_1. \end{cases} \tag{116}$$

Since the analysis of the Fokker-Planck equation is the same, we directly begin with the distribution function in Eq. (17) for $u_i = -w_i$ which is given by $P(|v|) \propto \delta(|v|)$. Namely, the only possible weights are $u_i = w_i = 0$, the same as the non-interpolation case. This is because the corresponding stationary distribution is

$$P(|v|) \propto \frac{1}{|v|^2(|v|+k)^2} \exp\left(-\frac{1}{2T}\int d|v|\frac{\beta_1(|v|+k) + \alpha_1\frac{1}{T}(|v|+k)^2}{\alpha_1|v|(|v|+k)^2}\right)$$
$$\propto |v|^{-\frac{3}{2}-\frac{\beta_1}{2T\alpha_1 k}}(|v|+k)^{-2+\frac{\beta_1}{2T\alpha_1 k}}. \tag{117}$$

The integral of Eq. (117) with respect to $|v|$ diverges at the origin due to the factor $|v|^{\frac{3}{2}+\frac{\beta_1}{2T\alpha_1 k}}$.

For the case $u_i = w_i$, the stationary distribution is given from Eq. (17) as

$$P(v) \propto \frac{1}{v^2(v-k)^2} \exp\left(-\frac{1}{2T}\int dv\frac{\beta_1(v-k) + \alpha_1 T(v-k)^2}{\alpha_1 v(v-k)^2}\right)$$
$$\propto v^{-\frac{3}{2}+\frac{\beta_1}{2T\alpha_1 k}}(v-k)^{-2-\frac{\beta_1}{2T\alpha_1 k}}. \tag{118}$$

Now, we consider the case of $\gamma \neq 0$. In the non-interpolation regime, when $u_i = -w_i$, the stationary distribution is still $p(v) = \delta(v)$. For the case of $u_i = w_i$, the stationary distribution is the same as in Eq. (17) after replacing $\beta$ with $\beta_2' = \beta_2 - \gamma$. It still has a phase transition. The weight decay has the effect of shifting $\beta_2$ by $-\gamma$. In the interpolation regime, the stationary distribution is still $p(v) = \delta(v)$ when $u_i = -w_i$. However, when $u_i = w_i$, the phase transition still exists since the stationary distribution is

$$p(v) \propto \frac{v^{-\frac{3}{2}+\theta_2}}{(v-k)^{2+\theta_2}} \exp\left(-\frac{\beta_1 \gamma}{2T\alpha_1} \frac{1}{k(k-v)}\right), \tag{119}$$

where $\theta_2 = \frac{1}{2T\alpha_1 k}(\beta_1 - \frac{\gamma}{k})$. The phase transition point is $\theta_2 = 1/2$, which is the same as the non-interpolation one.

The last situation is rather special, which happens when $\Delta = 0$ but $y \neq kx$: $y = kx - c/x$ for some $c \neq 0$. In this case, the parameters $\alpha$ and $\beta$ are the same as those given in Eq. (116) except for $\beta_2$:

$$\beta_2 = k\mathbb{E}[x^2] - kc = k\beta_1 - kc. \tag{120}$$

The corresponding stationary distribution is

$$P(|v|) \propto \frac{|v|^{-\frac{3}{2}-\phi_2}}{(|v|+k)^{2-\phi_2}} \exp\left(\frac{c}{2T\alpha_1} \frac{1}{k(k+|v|)}\right), \tag{121}$$

where $\phi_2 = \frac{1}{2T\alpha_1 k}(\beta_1 - c)$. Here, we see that the behavior of stationary distribution $P(|v|)$ is influenced by the sign of $c$. When $c < 0$, the integral of $P(|v|)$ diverges due to the factor $|v|^{-\frac{3}{2}-\phi_2} < |v|^{-3/2}$ and Eq. (121) becomes $\delta(|v|)$ again. However, when $c > 0$, the integral of $|v|$ may not diverge. The critical point is $\frac{3}{2} + \phi_2 = 1$ or equivalently: $c = \beta_1 + T\alpha_1 k$. This is because when $c < 0$, the data points are all distributed above the line $y = kx$. Hence, $u_i = -w_i$ can only give a trivial solution. However, if $c > 0$, there is the possibility to learn the negative slope $k$. When $0 < c < \beta_1 + T\alpha_1 k$, the integral of $P(|v|)$ still diverges and the distribution is equivalent to $\delta(|v|)$. Now, we consider the case of $u_i = w_i$. The stationary distribution is

$$P(|v|) \propto \frac{|v|^{-\frac{3}{2}+\phi_2}}{(|v|-k)^{2+\phi_2}} \exp\left(-\frac{c}{2T\alpha_1} \frac{1}{k-|v|}\right). \tag{122}$$

It also contains a critical point: $-\frac{3}{2} + \phi_2 = -1$, or equivalently, $c = \beta_1 - \alpha_1 kT$. There are two cases. When $c < 0$, the probability density only has support for $|v| > k$ since the gradient pulls the parameter $|v|$ to the region $|v| > k$. Hence, the divergence at $|v| = 0$ is of no effect. When $c > 0$, the probability density has support on $0 < |v| < k$ for the same reason. Therefore, if $\beta_1 > \alpha_1 kT$, there exists a critical point $c = \beta_1 - \alpha_1 kT$. When $c > \beta_1 - \alpha_1 kT$, the distribution function $P(|v|)$ becomes $\delta(|v|)$. When $c < \beta_1 - \alpha_1 kT$, the integral of the distribution function is finite for $0 < |v| < k$, indicating the learning of the neural network. If $\beta_1 \leq \alpha_1 kT$, there will be no criticality and $P(|v|)$ is equivalent to $\delta(|v|)$. The effect of having weight decay can be similarly analyzed, and the result can be systematically obtained if we replace $\beta_1$ with $\beta_1 + \gamma/k$ for the case $u_i = -w_i$ or replacing $\beta_1$ with $\beta_1 - \gamma/k$ for the case $u_i = w_i$.

## B    DERIVATION OF EQ. (14)

When $D = 0$, the Langevin equation (1) becomes

$$\begin{aligned}
dv &= -\nabla_v L + \sqrt{TC(v)} dW_t \\
&= -2(\beta_1' v - \beta_2) + \sqrt{4T(\alpha_1 v^2 - 2\alpha_2 v + \alpha_3)} dW_t,
\end{aligned} \tag{123}$$

where $\beta_1' := \beta_1 + \gamma$. According to the stationary distribution of the general SDE (27), by substituting $\mu(X)$ with $\beta_1' v - \beta_2$ and $B(X)$ with $\sqrt{4T(\alpha_1 v^2 - 2\alpha_2 v + \alpha_3)}$, we obtain

$$\begin{aligned}
P(v) &\propto \frac{1}{4T(\alpha_1 v^2 - 2\alpha_2 v + \alpha_3)} \exp\left[-\int dv \frac{4(\beta_1' v - \beta_2)}{4T(\alpha_1 v^2 - 2\alpha_2 v + \alpha_3)}\right] \\
&\propto (\alpha_1 v^2 - 2\alpha_2 v + \alpha_3)^{-1-\frac{\beta_1'}{2T\alpha_1}} \exp\left[-\frac{1}{T} \frac{\alpha_2 \beta_1' - \alpha_1 \beta_2}{\alpha_1 \sqrt{\Delta}} \arctan\left(\frac{\alpha_1 v - \alpha_2}{\sqrt{\Delta}}\right)\right],
\end{aligned} \tag{124}$$

where we utilize the integral

$$\int dv \frac{\beta_1' v - \beta_2}{\alpha_1 v^2 - 2\alpha_2 v + \alpha_3} = \frac{\alpha_2 \beta_1' - \alpha_1 \beta_2}{\alpha_1 \sqrt{\Delta}} \arctan\left(\frac{\alpha_1 v - \alpha_2}{\sqrt{\Delta}}\right) + \frac{\beta_1'}{2\alpha_1} \log(\alpha_1 v^2 - 2\alpha_2 + \alpha_3). \quad (125)$$

