# OpenReview forum: "Noise Balance and Stationary Distribution of Stochastic Gradient Descent"
_ICLR.cc/2025/Conference — ICLR 2025 Conference Withdrawn Submission_

### Official Review · Reviewer_VUhS · 2024-11-03

**Soundness:** 2
**Presentation:** 2
**Contribution:** 2
**Rating:** 3
**Confidence:** 3

**Summary:**

This work analyzes the SGD dynamics via the SDE approximation under the assumption of rescaling symmetry. The authors theoretically discover the "law of balance", indicating that SGD converges to a low-dimensional subspace with balanced weights. They further examine the stationary of SGD in a simplified diagonal linear network with 1-dimensional input and find various phenomena, including phase transitions, loss of ergodicity, and fluctuation inversion.

**Strengths:**

- The focus of this paper on SGD dynamics is essential in understanding the training process of deep neural networks. The authors prove the "law of balance" under the SDE approximation. The balancedness condition for SGD can potentially be novel compared to previous works. The condition helps to analyze the special case of diagonal linear nets with one-dimensional input (as a minimal model with concepts of width and depth), and the proof of the closed form of the stationary is technical and highly non-trivial (though I didn't carefully check all the proof).

- The toy model can exhibit rich phenomena like phase transitions or "fluctuation inversion". That means the analysis may have some practical implications since similar phenomena also happen in practice. I appreciate the authors' efforts in drawing connections between their theory and the practical phenomenons in the neural network training process, though there should not exist any overclaim (see weakness 2)

**Weaknesses:**

1. Does theorem 1 directly lead to balancedness? Notice that both $C_1$ and $C_2$ are functions of $u$ and $v$, and in (6) the matrices can be very ill-conditioned, which cannot directly imply the balance between $\|u\|$ and $\|w\|$. I believe in special cases like two-layer ReLU networks/diagonal linear nets, this can be correct as proved in the theorems. However, this paper states the "law of balance" as one of its main contributions. In the current form of this paper, the argument is limited to toy cases (both theory and experiments) and does not generalize to practical settings. To provide enough evidence, the authors should include more evidence (either theoretical or empirical) to verify the universality of this balance condition. For example, the authors could include more experiments with deep ReLU networks ($L\ge 3$) and check if the experiments also satisfy this balancedness. Otherwise, I suggest the authors put more focus on the theoretical results of the diagonal linear net and those empirical observations, and reconsider the paper's structure/message.

2. I don't think the edge of stability argument (Line 440-460) is valid. EoS is a phenomenon observed in full-batch GD cases without any gradient noise. However, the authors claim their deep model can explain the phenomenon by "the fluctuation of model parameters decreases as the gradient noise increases", which is clearly incorrect because there is no gradient noise for EoS with full batch GD. The authors should rethink what is the correct implication of this fluctuation inverse phenomenon instead of this argument.

3. Now the diagonal linear net only has 1-dimensional inputs, which is a minor limitation. I understand it could be too complicated to generalize to high-dimensional inputs so a minimal analytical model is more or less acceptable.

**Questions:**

Do the phenomena like phase transition/fluctuation inverse observed in the minimal model generalize to a larger/practical model?

---

### Official Review · Reviewer_2gby · 2024-11-04

**Soundness:** 2
**Presentation:** 3
**Contribution:** 2
**Rating:** 3
**Confidence:** 3

**Summary:**

This paper claims to analyze the noise effects in minibatch Stochastic Gradient Descent (SGD) using stochastic differential equations (SDE). This approach is intended to provide a theoretical framework for understanding the dynamics of minibatch SGD by modeling it as a continuous process.

**Strengths:**

In case it is applicable to mini batch SGD it provides great insights. But the authors should clarify this further.

**Weaknesses:**

The most relevant missing piece of information is a paragraph describing the long line of work about the fact that the implicit regularization effect of mini batch SGD is not explainable through means of continuous stochastic differential equations. Starting from Yaida (2018), continuing with Smith et al., "Origin of..." (2021), and Li et al., "What ..." (2022),  there has been a long line of works (40+) which deal with this problem. I believe discribing them and discussing the applicability to this setting it's absolutely necessary for a paper dealing exactly with this to be considered for publication.

See also Questions.

**Questions:**

The explicit claim in the abstract is that the noise of **minbatch SGD** is analyzed. However, the first formula oin the introduction (and the rest?) is about a stochastic differential equation. There has been a long line of works since 2019 showing that SGD and these SDEs have dynamics which are qualitatively very different, no matter the randomness utilized.
1) Does your result apply to mini batch SGD with a precise finite step size?
If the answer is no, then the paper has to be rewritten and I believe it is actively, on purpose, misleading the community. On top of this, if the answer is no, the results may not be novel at all.
If the answer is yes, then I propose reframing the paper because that is not clear.
2) If the answer is yes, under what assumptions on the data?
3) You describe what the result means in practice for linear networks. I believe that in the case considered, depending on how you sample batches, the result is some form of specific case of Damian et al. (2021) about label noise or Beneventano (2023) about random reshuffling. Can you please explain to me why that is/is not the case?
4) In case you are describing the discete dynamics of minbatch SGD, I'm curious if you can provide bounds on the speed of convergece and the learning rate.

My final grade and confidence will largely depend on the answers to these questions. I'm pretty sure about what I am writing, but I want to grant the authors with the benefit of the doubt.

---

### Official Review · Reviewer_hTqS · 2024-11-05

**Soundness:** 3
**Presentation:** 2
**Contribution:** 3
**Rating:** 6
**Confidence:** 3

**Summary:**

The paper explores the stationary distribution of the stochastic gradient descent (SGD) algorithm in the context of deep learning. This leads to the construction of a linear network model that captures network depth and width, providing an analytically tractable approach to understanding the stationary distribution of stochastic gradient flow. This distribution reveals complex phenomena, including phase transitions, ergodicity loss, memory effects, sign coherence, and fluctuation inversion. These findings underscore a key distinction between deep and shallow networks and offer theoretical insights relevant to variational Bayesian inference and the edge of stability in SGD.

**Strengths:**

The paper provides a detailed analysis of the stationary distribution of SGD in the training of multi-layer networks, yielding several novel insights into SGD dynamics.

The findings represent an improvement over prior work (e.g., Du 2018 and Wang 2022) and advance understanding of SGD’s role in deep networks.

The paper offers valuable implications connecting the results to edge of stability and Bayesian inference, presenting a well-rounded theoretical contribution.

**Weaknesses:**

The analysis is limited to a simplified “linear network” setting, which, while ideal for capturing rescaling symmetry, may not fully translate to practical networks where architectures involve skip connections and layer normalization layers. The applicability of these findings to state-of-the-art (SOTA) models might be limited, given that current deep learning architectures often deviate from the minimal settings examined in the paper.

Inconsistency of the arguments. In the equation (1), the update rule is SGD and $T=\eta/S$. In Section 4.4, the discussion of "infinite D limit" shows that $d/D *S/\eta=const$ will give $D\propto 1/\eta$. However, in the Figure 7, the network is trained with Adam whose scaling behavior is much different from that of SGD.

In terms of presentation, there are many temporary notations $\alpha_1, \alpha_2, \alpha_3$, ..., which are hard to understand how these parameters changes when varying the problem parameters.

**Questions:**

See weakness

---

### Official Review · Reviewer_EWjz · 2024-11-07

**Soundness:** 3
**Presentation:** 3
**Contribution:** 3
**Rating:** 6
**Confidence:** 5

**Summary:**

This papers studied SGD theoretically in simple settings, where rescaling sysmmetry exists. The authors reveal the "law of balance", i.e. the norm of each layer is balanced, and the gradient as well. Then for a special type of diagonal linear network, the stationary distribution is derived for different depth and width.

**Strengths:**

If I remembered correctly, I reviewed this paper twice already. The good thing is that the authors continuouly improved the paper in recent years.
-  This works studied a hard subject, i.e. the training dynamics of SGD, and sucessfully revealed some interesting aspects of SGD in the setting of rescaling symmetry.
- I appreciated the authors provide an analytical result of stationary distribution in diagonal linear network setting such that the effects of width and depth could be analysed clearly.
- The width/depth ratio is particularly interesting for making training stable.

**Weaknesses:**

The current analysis focused on the pure theoretical. It is extremely interesting and helpful to show that whether the insights obtained from such solid analysis could be applied to complex task settings and practical architectures. If this could be made from either more theoretical or empirical justification, it would a much stronger paper.  These apsects include whether stationary distribution could be obtained or not, whether the constant depth/width rule still holds or not, sign coherence happens or not, etc.

I think after many years of deep learning theory research, the entire ML community or industry call for a more insightful and practically useful results from theoretical understanding. This could benefit more and raise more impacts.

**Questions:**

See weakness.

---

### Official Review · Reviewer_1XTy · 2024-11-09

**Soundness:** 2
**Presentation:** 3
**Contribution:** 2
**Rating:** 3
**Confidence:** 2

**Summary:**

The paper at hand studies some symmetry phenomena. The authors start by studying the effects of symmetry on gradient dynamics and introduce their law of balance - stating that SGD dynamics obeys a specific law if some symmetry between weights is present. Then, the authors use this model to study SGD distribution and phase transition in the dynamics.

**Strengths:**

The topic of this paper is very interesting, and I like the authors approach. I just wish the paper was written in a different way (see below). The strengths are:

1) Claims are interesting. Even though some work on the topic (cited by the authors) exist, it is nice to revisit and propose a different perspective.

2) The paper dives directly into the subject, I like this.

3) Some empirical evidence complements claims.

**Weaknesses:**

This was an emergency review, so unfortunately, I do not have as much time as I wish to parse it. However, the main weakness I find is that the writing, claims, and flow are confusing. Theorems in the paper are opaque: setting, preciseness, and assumptions are often unclear. I looked at the appendix, and it did not help much.

I will place here my questions, which highlight weaknesses:

1) Theorem 3.1, especially in combination with Fig.1 is very confusing. To study the quantity you have, you use Ito's Lemma - and I see no problem with this. It is however highly unclear which noisy gradient model you choose (in the paper and even in the appendix). The usual model of gradient noise is additive Gaussian noise, and from what I understood in the appendix, you also use this. What is the difference between "SGD" and "Gaussian" in Figure 1? If you have an additive gaussian model, then these should be the same. If not, then you need a section to explain what you are doing!

2) Thm 3.1 Is stated as a differential equation driven by population quantities. I believe this equation only holds in expectation, correct? Even from the appendix, I find this hard to parse; I believe (related to point (1)) that you are claiming in some way that noisy components cancel? How can C1 and C2 then be expectation quantities?

3) Still about Thm 3.1. I find it very unclear how to interpret this: What does the equation converge to? Are C1 and S2 positive and definite? I guess it depends, and the equation alone cannot predict if $u$ and $v$ get close together. Yet I sense that this is later assumed in further statements.

4) Thm 4.1: the proof of this statement is very confusing: I cannot track what the authors assumed from the appendix. Do you presume balance? I am sorry that I could not get much out of this.

5) From Section 4.3 I got lost. I agree that formula (15) is potentially interesting, but there are so many open points left that I could not continue confidently.

**Questions:**

See above.

---

### Author Response · Authors · 2024-11-30
**Rebuttal and thanks**

We would like to thank all the reviewers for the detailed and constructive feedback. We learned a lot from the review and will update our manuscript accordingly in the future. After careful thought, we decided to withdraw the submission.

---

### Note · Authors · 2024-11-30

I have read and agree with the venue's withdrawal policy on behalf of myself and my co-authors.